# Structural engineering of chimeric antigen receptors targeting HLA-restricted neoantigens

Michael S. Hwang[1,2,3,14,17], Michelle S. Miller [2,4,5,15,17], Puchong Thirawatananond[4], Jacqueline Douglass [1,2,3], Katharine M. Wright [2,4,5], Emily Han-Chung Hsiue[1,2,3], Brian J. Mog [1,2,3,6], Tihitina Y. Aytenfisu[4], Michael B. Murphy[7], P. Aitana Azurmendi [4], Andrew D. Skora[1,2,16], Alexander H. Pearlman [1,2,3], Suman Paul [1,2,3,8], Sarah R. DiNapoli [1,2,3], Maximilian F. Konig [1,2,3,9], Chetan Bettegowda[1,3,8,10], Drew M. Pardoll[5,8], Nickolas Papadopoulos [1,3,8,11,12], Kenneth W. Kinzler[1,3,5,8,12], Bert Vogelstein [1,2,3,5,8,11,12✉], Shibin Zhou [1,3,5,8✉] & Sandra B. Gabelli [4,8,13✉]

Chimeric antigen receptor (CAR) T cells have emerged as a promising class of therapeutic agents, generating remarkable responses in the clinic for a subset of human cancers. One major challenge precluding the wider implementation of CAR therapy is the paucity of tumor-specific antigens. Here, we describe the development of a CAR targeting the tumor-specific isocitrate dehydrogenase 2 (IDH2) with R140Q mutation presented on the cell surface in complex with a common human leukocyte antigen allele, HLA-B*07:02. Engineering of the hinge domain of the CAR, as well as crystal structure-guided optimization of the IDH2$^{R140Q}$-HLA-B*07:02-targeting moiety, enhances the sensitivity and specificity of CARs to enable targeting of this HLA-restricted neoantigen. This approach thus holds promise for the development and optimization of immunotherapies specific to other cancer driver mutations that are difficult to target by conventional means.

[1] Ludwig Center, Sidney Kimmel Comprehensive Cancer Center, Johns Hopkins University School of Medicine, Baltimore, MD, USA. [2] Howard Hughes Medical Institute, Chevy Chase, MD, USA. [3] Lustgarten Laboratory for Pancreatic Cancer Research, Sidney Kimmel Comprehensive Cancer Center, Johns Hopkins University School of Medicine, Baltimore, MD, USA. [4] Department of Biophysics and Biophysical Chemistry, Johns Hopkins University School of Medicine, Baltimore, MD, USA. [5] Bloomberg~Kimmel Institute for Cancer Immunotherapy, Sidney Kimmel Comprehensive Cancer Center, Baltimore, MD, USA. [6] Department of Biomedical Engineering, Johns Hopkins University, Baltimore, MD, USA. [7] Cytiva, Marlborough, MA, USA. [8] Department of Oncology, Johns Hopkins University School of Medicine, Baltimore, MD, USA. [9] Division of Rheumatology, Department of Medicine, Johns Hopkins University School of Medicine, Baltimore, MD, USA. [10] Department of Neurosurgery, Johns Hopkins University School of Medicine, Baltimore, MD, USA. [11] Department of Pathology, Johns Hopkins University School of Medicine, Baltimore, MD, USA. [12] Sol Goldman Pancreatic Cancer Research Center, Johns Hopkins University School of Medicine, Baltimore, MD, USA. [13] Department of Medicine, Johns Hopkins University School of Medicine, Baltimore, MD, USA. [14] Present address: Genentech, Inc., South San Francisco, CA, USA. [15] Present address: Walter and Eliza Hall Institute of Medical Research, Parkville, VIC, Australia. [16] Present address: Lilly Biotechnology Center, Eli Lilly and Co, San Diego, CA, USA. [17] These authors contributed equally: Michael S. Hwang, Michelle S. Miller. ✉email: vogelbe@jhmi.edu; sbzhou@jhmi.edu; gabelli@jhmi.edu

C himeric antigen receptor (CAR) T cells are an emerging class of therapeutic agents that have induced impressive responses and major clinical benefit in patients with B-cell malignancies[1–6]. CD19 is expressed across the B-cell lineage and, despite concomitant depletion of normal B cells, CD19-targeting CAR therapy has manageable adverse effects and is generally well tolerated by patients. The majority of other CAR T cells developed to date have also been directed against tumor-associated antigens (TAAs)[7]. Although TAAs are typically overexpressed on tumors, CAR T cells targeting TAAs can result in "on-target, off-tumor" toxicities when such antigens are also expressed by non-neoplastic cells. Such TAA toxicity profiles have proven to be unacceptable and sometimes fatal[8–12]. These experiences highlight the need to identify antigens that are exclusively expressed in cancer cells to promote the wider implementation of CAR therapies.

In the absence of truly tumor-specific antigens for CAR therapy, methods employing combinatorial antigen targeting have been developed to reduce off-tumor toxicity[13–17]. For example, although efforts by Perna et al. to identify singular CAR targets for acute myeloid leukemia (AML) were unable to reveal candidates with a profile as favorable as CD19, a generalizable approach and discovery platform for combinatorial antigen targeting was described[18]. Despite the advent of combinatorial antigen targeting, the lack of tumor-specific antigens will continue to impede the translation of powerful immunotherapeutic agents, such as CAR T cells, to the clinic[19].

Genetic alterations that drive tumorigenesis can result in tumor-specific vulnerabilities that can be exploited therapeutically[20,21]. However, therapies targeting alterations such as those arising from hotspot mutations of cancer driver oncogenes are limited[22]. The majority of proteins encoded by commonly mutated cancer driver genes are intracellular, which precludes their targeting by standard antibody-based biologics[21]. One intracellular protein that is frequently mutated in cancer is isocitrate dehydrogenase 2 (IDH2), somatic mutations of which can contribute to the development of AML via production of the oncometabolite 2-hydroxyglutarate (2HG)[23,24]. Oncogenic driver mutations of *IDH2* occur in 15–25% of AML patients and almost exclusively at the arginine residues R140 and R172 with the most common being R140Q[25–28]. *IDH2* mutations can also be detected in cholangiocarcinoma, chondrosarcoma, and glioma[29].

Mutated protein products derived from cancer driver genes, or mutation-associated neoantigens (MANAs), can be processed and presented as peptides by human leukocyte antigen (HLA) molecules[30–32]. Such altered peptides can form peptide-HLA (pHLA) complexes resulting in tumor-specific antigens that can be engaged at the cell surface[33–38]. Recently, T-cell receptor-mimic (TCRm) antibodies have been generated that can discern subtle differences between pHLA complexes derived from the protein products of cancer driver hotspot mutations (i.e., MANAs) and those from their wild-type (WT) counterparts[39,40]. Although TCRm antibodies have been converted to CAR T-cell therapeutic formats, all but one target HLA complexed with wild-type peptides derived from TAAs rather than HLA complexed with mutant peptides derived from cancer driver genes[41–47].

Structural analyses of TCR and TCRm antibodies have revealed distinct binding characteristics. For example, crystal structures of TCR-pHLA complexes generally favor a canonical diagonal binding mode, where the TCR Vα complementarity determining regions (CDRs) sit atop the N-terminus of the peptide and the Vβ CDRs over the C-terminus[48,49]. The growing number of TCRm antibody structures bound to pHLA, however, are beginning to reveal a wide range of orientations, as measured by docking and incident angles[48]. A comparison of the available TCRm/pHLA complex structures has failed to display a correlation between the

binding orientation and the binding strength or mechanism. Examples include the ESK1 antigen-binding fragment (Fab) which only makes contacts with the N-terminus of the Wilms tumor protein 1 (WT1) peptide[50] and the Hyb3 Fab in complex with melanoma-associated antigen (MAGE)-pHLA-A1, where the Fab only interacts with the C-terminus of MAGE[51]. As such, a greater structural understanding of TCRm binding could help guide the design of more efficacious and safer TCRm antibodies, including those in a CAR T-cell format.

Here, we describe the identification of a neoantigen-specific TCRm antibody targeting the IDH2[R140Q] mutant peptide in complex with a common HLA allele (HLA-B*07:02), the structural basis of its specificity, and its functional activity in a CAR T-cell format.

## Results

**Generation of a CAR T-cell design targeting HLA-B*07:02-restricted IDH2[R140Q] peptide.** Both the mutant IDH2[R140Q] (aa 134–143, SPNGTIQNIL) and wild-type IDH2[WT] (SPNGTIRNIL) peptides are presented as pHLA complexes on HLA-B*07:02[52,53]. NetMHCpan, which predicts peptide binding to HLA alleles, estimates that the mutant IDH2[R140Q] peptide will bind 200-fold tighter to the HLA-B*07:02 allomorph compared with other common HLA alleles. To identify TCRm antibodies that could selectively target the IDH2[R140Q]-pHLA, two combinatorial phage libraries displaying single-chain variable fragments (scFvs) were screened for binders that could selectively target mutant IDH2[R140Q]-HLA-B*07:02 but not wild-type IDH2[WT]-HLA-B*07:02[39,40]. A phage display approach was employed to not only allow for facile grafting to multiple therapeutic platforms, but also to enable protocol customization in favor of stringent negative selection. Positive selection was conducted with HLA-B*07:02 pHLA monomers containing the IDH2[R140Q] peptide, and negative selection was performed against pHLA monomers containing the IDH2[WT] peptide. Selected phage clones were amplified and assessed for their ability to bind to RPMI-6666 cells, which endogenously express HLA-B*07:02, presenting the mutant or WT peptide via flow cytometry. Five phage clones were obtained that selectively bound IDH2[R140Q] but not IDH2[WT] (Supplementary Fig. 1, Supplementary Table 1).

In designing a CAR capable of targeting IDH2[R140Q]-HLA-B*07:02, we focused on the extracellular components of the CAR molecule—the binding moiety (here, an scFv) and the hinge region of the CAR. The base CAR design employed a 3rd generation construct, comprising of a CD28 transmembrane domain and two intracellular domains, the 4-1BB and CD3ζ signaling domains (Fig. 1a). Customization of the hinge/spacer region of CARs has previously been explored as a means to improve therapeutic activity[54–61]. As such, prior to evaluating the five IDH2[R140Q] neoepitope-specific scFvs in the CAR T-cell format, we first sought to optimize the hinge region of the CAR. Four hinge regions that have been previously utilized in other CAR designs were screened: one derived from CD8α, two derived from CD28 of varied lengths, and one derived from immunoglobulin gamma 1 (IgG1, mutFc) (Fig. 1a). Both the CD8α and the short CD28 (CD28s) hinges were derived from their respective receptor's membrane-proximal stalk regions, up to their Ig-like V-type domains. The long CD28 hinge (CD28l) was similar to the CD28s hinge but included a portion of the Ig-like V-type domain up to the membrane-proximal cysteine residue responsible for intrachain disulfide bonding. The mutant fragment crystallizable region (mutFc) hinge was derived from the hinge, constant heavy chain 2 ($C_H2$), and constant heavy chain 3 ($C_H3$) regions of IgG1 Fc. Mutations were incorporated in the PELLGG and ISR motifs to abrogate its interaction with the IgG1-Fc gamma receptor

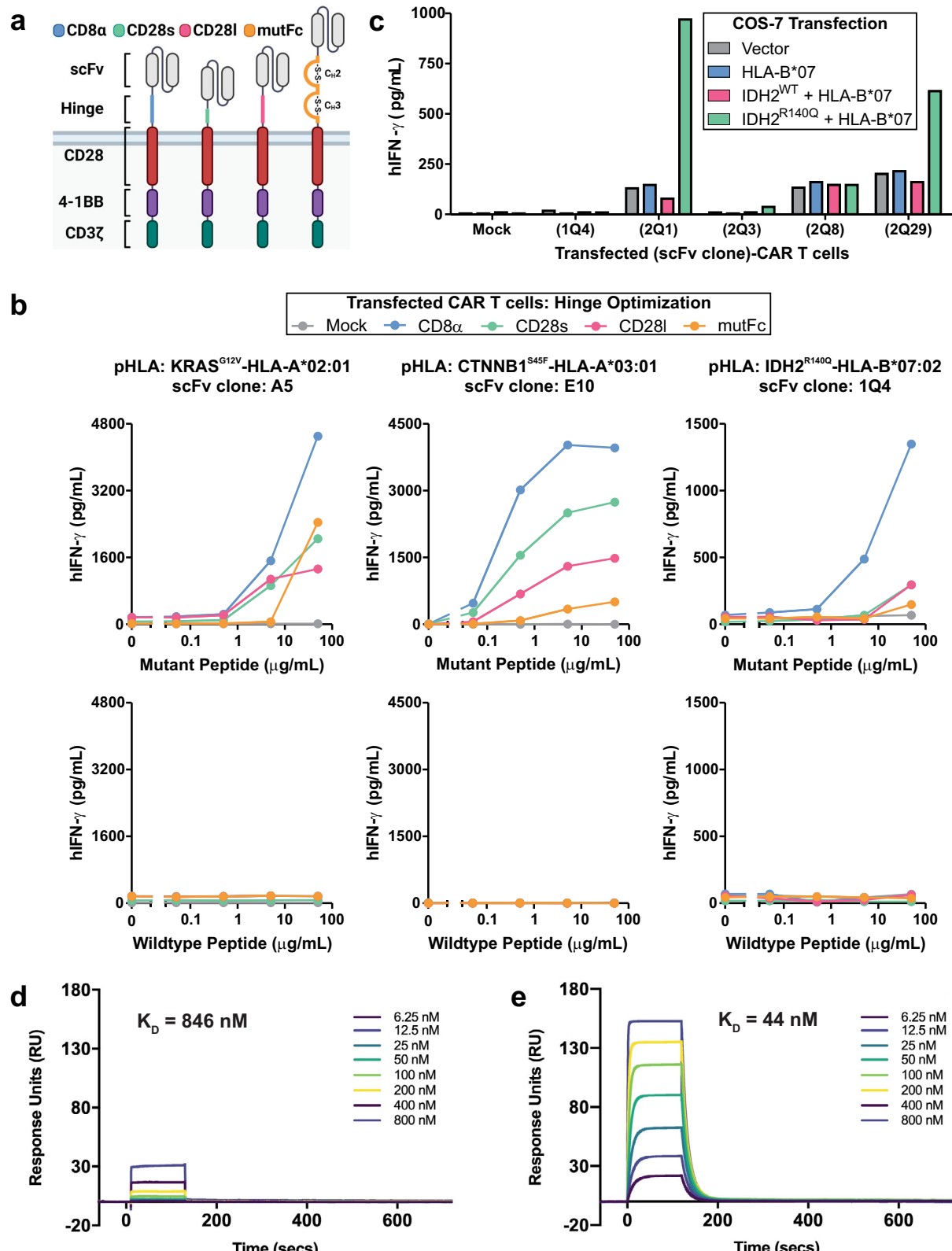

(FcγR) and reduce off-target CAR activation[62]. To determine whether a particular hinge would promote improved targeting properties across multiple HLA-restricted MANA targets, the four CAR hinges were tested in combination with three TCRm scFvs each targeting a unique mutant pHLA complex from three common HLA alleles: KRAS$^{G12V}$-HLA-A*02:01, β-catenin CTNNB1$^{S45F}$-HLA-A*03:01, and IDH2$^{R140Q}$-HLA-B*07:02[39,40].

An mRNA-based expression system was employed to facilitate rapid testing of candidate CAR constructs in primary human T cells. Optimization of the system allowed for near 100% electroporation efficiency while retaining high cell viability (Supplementary Fig. 2). Reactivity of T cells engineered with the above CAR constructs was assessed by interferon-γ (IFN-γ) release after co-incubation with target cells loaded with the

**Fig. 1 Hinge region optimization of HLA-restricted MANA-targeting CARs. a** Illustration summarizing the CAR designs employed in this study. Colors indicating hinge region variations: CD8α, blue; CD28s (short CD28), green; CD28l (long CD28), pink; mutFc (mutant Fc), orange. **b** T cells harboring each of the three MANA-targeting CARs, with the indicated hinge regions, were co-incubated with target cells loaded with a titration of the corresponding mutant and WT peptides. The following target cells were employed: T2 cells expressing endogenous HLA-A*02:01 for *KRAS*, T2A3 cells overexpressing HLA-A*03:01 for *CTNNB1*, and HLA-B*07:02-transfected COS-7 cells for *IDH2*. T-cell activation, as assessed by IFN-γ release, was measured by ELISA. Data are representative of three independent experiments. **c** T cells harboring IDH2$^{R140Q}$-HLA-B*07:02-targeting scFvs, grafted onto a CD8α-hinged CAR, were co-incubated with COS-7 cells transfected with *HLA-B*07:02* and either full-length *IDH2$^{R140Q}$* or *IDH2$^{WT}$*. T-cell activation, as assessed by IFN-γ release, was measured by ELISA. Data are representative of three independent experiments. **d** 2Q1-Fab binding to IDH2$^{WT}$-HLA-B*07:02 was measured by multi-cycle kinetics using SPR with increasing concentrations (6.25, 12.5, 25, 50, 100, 200, 400, 800 nM) of purified 2Q1-Fab ($N = 2$). The curves were fit with a 1:1 binding model to calculate the listed $K_D$. **e** Same as (**d**) using IDH2$^{R140Q}$-HLA-B*07:02. Source data are provided as a Source data file.

corresponding mutant and WT peptides. Across all three target HLA alleles tested, the CD8α hinge provided the highest signal across the spectrum of mutant-peptide concentrations and thus was selected as the preferred hinge design (Fig. 1b).

Five scFvs (1Q4, 2Q1, 2Q3, 2Q8, 2Q29) targeting IDH2$^{R140Q}$-HLA-B*07:02 were grafted onto the CD8α hinge-optimized CAR construct and expressed in primary human T cells. When co-incubated with COS-7 cells co-transfected with plasmids encoding HLA-B*07:02 and either full-length IDH2$^{R140Q}$ or IDH2$^{WT}$, T cells harboring the 2Q1 CAR showed the most robust activation signal in response to IDH2$^{R140Q}$, as assessed by IFN-γ release. In addition, these T cells showed the lowest reactivity to IDH2$^{WT}$ (Fig. 1c). Therefore, the 2Q1 CAR was selected as the lead candidate. It should be noted that the CD8α hinge-optimized 2Q1 CAR is highly sensitive to the IDH2$^{R140Q}$ neoantigen, as mass spectrometry data from transfected COS-7 cells indicated that IDH2$^{R140Q}$ antigen density was ~25 copies per cell[53]. The specificity of clone 2Q1 was further assessed by surface plasmon resonance (SPR). At high concentrations, there was detectable binding of clone 2Q1 to IDH2$^{WT}$-HLA-B*07:02 with a calculated $K_D > 800$ nM (Fig. 1d). In contrast, clone 2Q1 bound more tightly to IDH2$^{R140Q}$-HLA-B*07:02 with a $K_D$ of 44 nM, a $k_{on}$ of $1.59 \times 10^6$ M$^{-1}$ s$^{-1}$, and a $k_{off}$ of $9.66 \times 10^{-2}$ s$^{-1}$ (Fig. 1e, Supplementary Table 2).

**The IDH2$^{R140Q}$ epitope is buried in the peptide-HLA-B*07:02 binding cleft.** To investigate the structural basis of 2Q1 selectivity, we determined the structures of the peptide-bound HLA-B*07:02 (Fig. 2a–e, Table 1). The IDH2$^{WT}$ and IDH2$^{R140Q}$ peptides bound in essentially identical conformations, with alignment of the peptide Cα atoms yielding a root-mean-square deviation (rmsd) of 0.193 Å (Fig. 2f). The only difference between the two HLA-bound peptides was at the site of mutation. Surprisingly, the epitope residue of interest, IDH2$^{140}$, is buried deep within the peptide-binding groove of the HLA-B*07:02 structure, in contrast to other MANA-pHLA targets (Fig. 2b, d, e)[40,49,63]. In the IDH2$^{WT}$ peptide, IDH2$^{R140}$ was held in place by residues located in the β-sheet floor and α2 helix of HLA-B*07:02. Specifically, IDH2$^{R140}$ was secured via a salt-bridge with Asp114 (β-sheet), hydrogen bonds to Arg156 (α2) and Tyr116 (β-sheet), and π-π interactions with Tyr99 (β-sheet) (Fig. 2f). In the IDH2$^{R140Q}$ peptide, IDH2$^{R140Q}$ also interacts with Arg156 and Tyr116, but the shorter side-chain length precluded interaction with Asp114 (Fig. 2f). Furthermore, substitution of IDH2$^{R140}$ with glutamine abrogates π-π interactions with Tyr99. This loss of interaction results in a 5° decrease in the melting temperature of the complex (Supplementary Fig. 3).

**Structural determinants of 2Q1 selectivity for IDH2$^{R140Q}$-HLA-B*07:02.** The remarkable similarity in peptide-bound conformations, along with the buried epitope residue, IDH2$^{R140Q}$, raised the question of how 2Q1 was able to selectively

bind the mutant pHLA (Fig. 1). Attempts to crystallize the scFv-pHLA complex were unsuccessful, so the 2Q1 scFv sequence was grafted into an IgG1 format and expressed as a full-length antibody. The 2Q1-Fab fragment was purified following papain cleavage. The crystal structure of the purified 2Q1-Fab/IDH2$^{R140Q}$-HLA-B*07:02 ternary complex (Supplementary Fig. 4a, b, c, d) was refined to 2.9 Å resolution with one complex molecule in the asymmetric unit (Fig. 3a, Table 1). Despite the limited resolution of the data, clear electron density was observed at the binding interface for the IDH2$^{R140Q}$ peptide and the CDRs of the 2Q1-Fab (Supplementary Fig. 4e). The 2Q1-Fab docks onto IDH2$^{R140Q}$-HLA-B*07:02 in a parallel orientation with a docking angle of 20° (Fig. 3b) and was shifted towards the C-terminus of the IDH2$^{R140Q}$ peptide with an incident angle of 19°. The total buried surface area of the 2Q1-Fab/IDH2$^{R140Q}$-HLA-B*07:02 interface was 1264 Å$^2$, with the heavy chain contributing twice as much as the light chain (802 and 462 Å$^2$, respectively) (Supplementary Table 3). Overall, the 2Q1-Fab made a total of 43 contacts with the IDH2$^{R140Q}$ peptide, mediated primarily by CDR-L1 and CDR-H3. Notably, IDH2$^{R140Q}$ makes one direct contact with the 2Q1-Fab, with the backbone carbonyl forming a hydrogen bond with the side chain of Arg102 (CDR-H3) (Fig. 3a, c). All the other contacts are made with neighboring residues on the IDH2$^{R140Q}$ peptide. Specifically, the backbone of IDH2$^{G137}$ is within hydrogen bond distance of Asn30 (CDR-L1), while IDH2$^{I139}$ sits inside a 'pocket' at the 2Q1-Fab interface that is formed by Asn30, Thr31, Ala32 (CDR-L1), Tyr92 (CDR-L3), Trp101, and Arg102 (CDR-H3) (Fig. 3a, c, d). Furthermore, the carboxamide side chain of IDH2$^{N141}$ forms a hydrogen bond with the backbone nitrogen of Arg102 (CDR-H3). In contrast, recognition of HLA-B*07:02 is mediated by all six CDRs (Fig. 3a). There are a total of 116 contacts between the 2Q1-Fab and the peptide-binding cleft alpha helices, with each chain contributing equally.

The determinants of selectivity were not immediately obvious from these contacts. Structural alignment of the HLA-B*07:02 of the 2Q1-Fab bound and unbound IDH2$^{R140Q}$-pHLA structures showed no large conformational changes in the peptide-binding groove (Supplementary Fig. 5a, rmsd 0.66 Å over 277 Cα of HLA-B*07:02). However, superposition of the 2Q1-Fab bound IDH2$^{R140Q}$-pHLA and IDH2$^{WT}$-pHLA structures revealed a slight shift in the backbone carbonyl of IDH2$^{R140Q}$, which is the only direct interaction between that residue and 2Q1-Fab (Fig. 3d, Supplementary Fig. 5b). We hypothesize that the charge-assisted hydrogen bond formed between Arg102 (CDR-H3) in the 2Q1-Fab and the backbone carbonyl of IDH2$^{R140Q}$ is key to its selectivity (Fig. 3a, c, d, Supplementary Fig. 5b). The longer side chain of the WT arginine residue, IDH2$^{R140}$, is buried deep in the peptide-binding groove and forms an additional charged interaction with Asp114 (HLA), along with π-π interactions with Tyr99 (HLA), that IDH2$^{R140Q}$ cannot form (Fig. 2f). This creates greater flexibility in the mutant-peptide backbone, allowing a subtle shift in the conformation of the peptide backbone induced

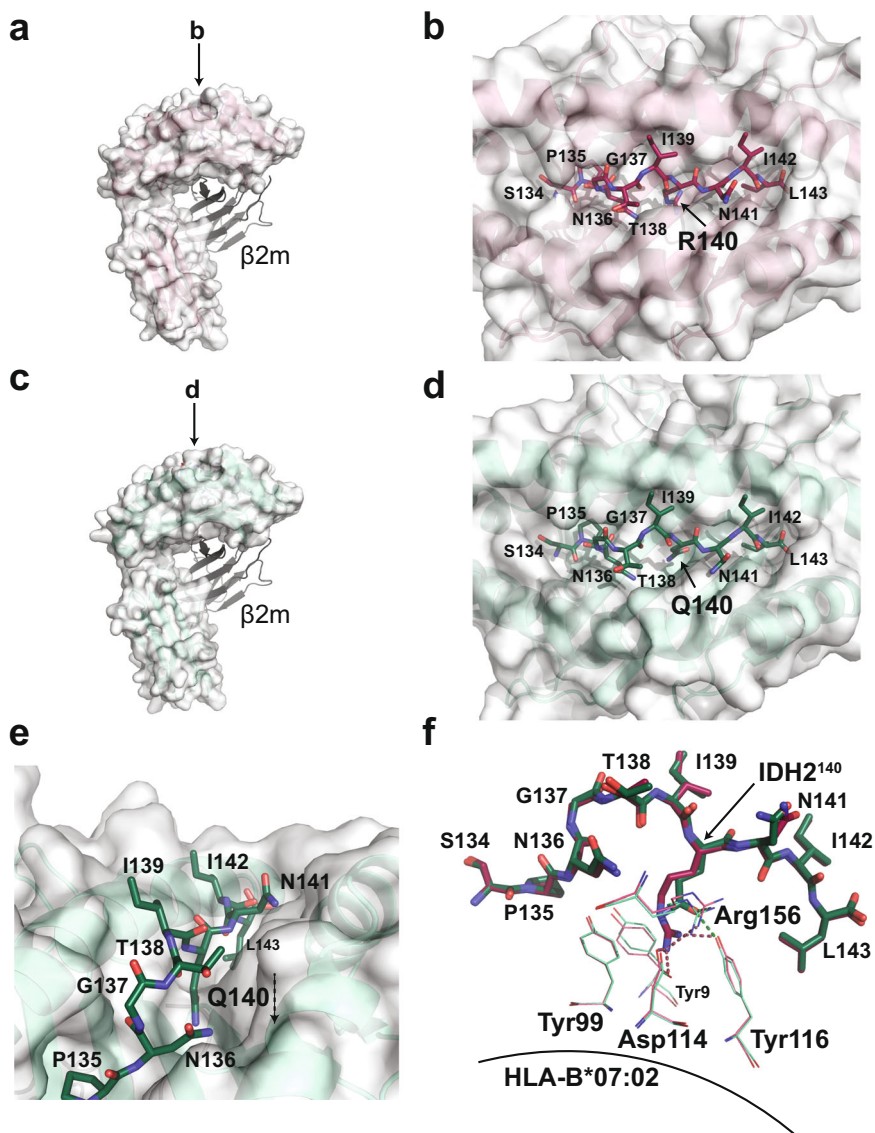

**Fig. 2 The IDH2<sup>R140Q</sup> epitope is buried in the peptide-HLA-B*07:02 complex. a** Overall structure of the IDH2<sup>WT</sup> peptide bound to HLA-B*07:02 (PDB ID 6UJ8). The IDH2<sup>WT</sup>-HLA-B*07:02 complex is formed as a heterodimer between the heavy chain, represented as a white surface, and β2-microglobulin, represented as cartoon. The resolution of the IDH2<sup>WT</sup>-HLA-B*07:02 structure is 2.25 Å with an average B-factor of 40 Å². **b** A bird's-eye view of the IDH2<sup>WT</sup> peptide sitting in the HLA-B*07:02 peptide-binding groove. Amino acid residues of the IDH2<sup>WT</sup> peptide are labeled. **c** Overall structure of IDH2<sup>R140Q</sup> peptide bound to HLA-B*07:02 (PDB ID 6UJ7). The heavy chain is represented as a white surface, and β2-microglobulin, represented as cartoon. The resolution of the IDH2<sup>R140Q</sup>-HLA-B*07:02 structure is 1.9 Å with an average B-factor of 34 Å². **d** A bird's-eye view of the IDH2<sup>R140Q</sup> peptide bound in the HLA-B*07:02 binding pocket. The residue of interest, IDH2<sup>R140Q</sup>, is buried in the peptide-binding groove. **e** Different orientation of (**d**) highlighting the downward position of the epitope of interest. **f** Structural alignment of the WT and mutant IDH2 peptides, showing the interactions between the epitope residue of interest and residues on the β-sheet floor of HLA-B*07:02. IDH2 peptide residues are labeled with single-letter amino acid codes. The HLA-B*07:02 residues are shown in green (mutant) and raspberry (WT) lines and labeled with three-letter amino acid codes; hydrogen bonds are represented as dotted lines.

by Fab binding and the formation of a hydrogen bond between Arg102 (CDR-H3) and the peptide (Fig. 3a, d, Supplementary Fig. 5b). While possibly still able to form in the WT pHLA complex, this subtle conformational shift is likely to be more energetically favorable in the mutant pHLA complex. The 1.2 Å shift in the coordinates of the carbonyl oxygen atoms between IDH2<sup>R140Q</sup> in the ternary complex (PDB ID 6UJ9, 2.9 Å) and IDH2<sup>WT</sup> in the WT pHLA structure (PDB ID 6UJ8, 2.25 Å), albeit small, is about three-times the estimated coordinate error of 0.41 and 0.2 Å, respectively. To test the importance of this hydrogen bond for affinity, we mutated Arg102 (CDR-H3) to an alanine residue (R102A) and evaluated the binding of this clone

(2Q1.1) to both the WT and mutant pHLA-B*07:02 by enzyme-linked immunosorbent assay (ELISA) (Fig. 3e). We found that the R102A mutation completely abrogated binding to both WT and mutant pHLA, confirming the importance of this hydrogen bond to the overall affinity and interaction of 2Q1 with IDH2<sup>R140Q</sup>-HLA-B*07:02.

To deepen the understanding of 2Q1 recognition, we aligned and performed subsequent structural analysis of the HLA-B allomorphs, focusing on polymorphic residues that could affect peptide-MHC binding and recognition by the TCRm antibody. For example, buried Tyr116 in HLA-B*07:02 (Asp116 in HLA-B*27:01, Phe116 in HLA-B*35:01) which is at the β-sheet base, is

**Table 1 Data collection and refinement statistics.**

| | IDH2^WT^-HLA-B*07:02 (PDB ID 6UJ8) | IDH2^R140Q^-HLA-B*07:02 (PDB ID 6UJ7) | 2Q1-Fab (PDB ID 7KGU) | 2Q1-Fab/IDH2^R140Q^-HLA-B*07:02 (PDB ID 6UJ9) |
|---|---|---|---|---|
| *Data collection* | | | | |
| Diffraction source | NSLS-II X17-ID-2 | NSLS-II X17-ID-1 | NSLS-II X17-ID-1 | NLSL-II X17-ID-1 |
| Wavelength (Å) | 0.9793 | 0.9996 | 0.9201 | 0.9201 |
| Temperature (K) | 100 | 100 | 100 | 100 |
| Detector | Dectris EIGER X 16M | Dectris EIGER X 9M | Dectris EIGER X 9M | Dectris EIGER X 9M |
| Rotation range per image (°) | 0.2 | 0.1 | 0.2 | 0.5 |
| Total rotation range (°) | 268 | 320 | 143 | 160 |
| Space group | p2$_1$ | p2$_1$ | p2$_1$ | p2$_1$ |
| $a$, $b$, $c$ (Å) | 67.69, 70.48, 88.15 | 67.04, 70.67, 87.48 | 38.46, 263.95, 91.24 | 83.61, 42.01, 125.18 |
| $\alpha$, $\beta$, $\gamma$ (°) | 90.00, 107.65, 90.00 | 90.00, 107.4, 90.00 | 90.00, 99.53, 90.00 | 90.00, 92.75, 90.00 |
| Resolution range (Å) | 47.58–2.25 (2.31–2.25) | 29.15–1.90 (1.95–1.90) | 47.58–2.40 (2.46–2.40) | 47.58–2.90 (2.98–2.90) |
| Total no. of observations | 191,270 | 365,739 | 197,009 | 60,383 |
| No. of unique observations | 37,763 | 61,233 | 68,740 | 19,451 |
| Completeness (%) | 99.7 (99.7) | 99.2 (92.2) | 98.7 (98.8) | 98.1 (96.4) |
| Redundancy | 5.0 (4.6) | 6.0 (4.2) | 2.8 (2.6) | 3.1 (3.0) |
| $\langle I/\sigma(I)\rangle$ | 10.4 (2.1) | 13.6 (2.0) | 8.8 (2.1) | 6.5 (2.0) |
| $R_{merge}$ | 0.096 (0.65) | 0.068 (0.63) | 0.096 (0.46) | 0.150 (0.56) |
| $CC_{1/2}$ | 0.99 (0.76) | 0.99 (0.73) | 0.99 (0.67) | 0.98 (0.71) |
| *Refinement* | | | | |
| Resolution range (Å) | 47.58–2.25 (2.31–2.25) | 47.43–1.90 (1.95–1.90) | 45.53–2.40 (2.46–2.40) | 48.93–2.90 (2.97–2.90) |
| No. of reflections, working set | 35,786 | 58,162 | 65,303 | 18,480 |
| No. of reflections, test set | 1884 | 3062 | 3437 | 973 |
| $R_{work}/R_{free}$ | 0.21/0.26 (0.31/0.38) | 0.18/0.23 (0.29/0.32) | 0.20/0.26 (0.31/0.37) | 0.21/0.28 (0.30/0.39) |
| No. of non-H atoms | | | | |
| Protein | 6,354 | 6,401 | 12,904 | 6,460 |
| Ligand/ion | 96 | 9 | 257 | 95 |
| Water | 170 | 496 | 595 | 2 |
| R.m.s. deviations | | | | |
| Bonds (Å) | 0.008 | 0.009 | 0.012 | 0.007 |
| Angles (°) | 1.53 | 1.57 | 1.81 | 1.54 |
| Average B factors (Å$^2$) | | | | |
| Protein | 44.8 | 37.5 | 42.2 | 43.9 |
| Ligand/ion | 66.9 | 69.9 | 67.7 | 85.7 |
| Water | 42.6 | 40.7 | 35.5 | 28.9 |
| Ramachandran (%) | | | | |
| Favorable | 97.0 | 95.8 | 96.0 | 94.8 |
| Allowed | 2.8 | 3.9 | 4.0 | 4.8 |
| Disallowed | 0.2 | 0.3 | 0 | 0.4 |

at hydrogen bonding distance of IDH2^R140Q^, the neoantigen displayed. This would significantly affect binding of the neoantigen to the HLA. Another polymorphic residue worth mentioning is Asn63 in HLA-B*07:02 (Glu63 in HLA-B*27:01 and HLA-B*13:01) since the glutamate polymorphism would clash with the proline anchor residue at position 2 of the IDH2^R140Q^ peptide[64].

**2Q1-Fab undergoes a conformational change upon binding its target IDH2^R140Q^-HLA-B*07:02.** To assess if there was conformational change in the 2Q1-Fab upon pHLA binding, we determined the structure of the 2Q1-Fab in free form. There were four copies of the 2Q1-Fab in the asymmetric unit with alignment of the peptide Cα atoms yielding pairwise rmsd values of 0.25–0.32 Å. Overall, the structures of the four copies are similar, with the only striking difference being the conformation of CDR-H3 (Supplementary Fig. 6). This is likely a result of the conformational flexibility inherent in this loop. Structural alignment of the Cα atoms of the unbound 2Q1-Fab molecule with the IDH2^R140Q^-HLA-B*07:02 bound 2Q1-Fab display pairwise rmsd values between 1.2 and 1.4 Å. This superposition revealed a rmsd of 1.3 Å displacement (maximum rmsd of 3.6 Å) in the light

chain, while the outer strands of the β-sheet of the heavy variable chain expand to accommodate binding to the HLA-B*07:02 with an rmsd of 1.5 Å (maximum rmsd of 7.4 Å) (Supplementary Fig. 6). All three CDRs of the 2Q1 heavy chain exhibited significant conformational changes (Fig. 4a). Specifically, binding to the C-terminal end of HLA-B*07:02 α1 and the N-terminal end of α2 induces conformational changes in CDR-H1 and -H2, respectively, while binding to the C-terminal end of the IDH2 peptide induces a conformational shift in CDR-H3, thus optimizing binding to pHLA (Fig. 4a, b).

**Structure-guided optimization of 2Q1 targeting moiety.** In order to improve the selectivity of 2Q1, we mutated—in silico—all CDR residues within 8 Å of the epitope residue of interest, IDH2^R140Q^, to each of the other 19 naturally occurring amino acids and calculated the predicted change in affinity for the mutant pHLA. Eight single amino acid substitutions targeting CDR-H3 (amino acids W101, Y103) and CDR-L1 (amino acid A32) were selected for experimental evaluation based on their predicted binding affinities (Supplementary Fig. 7a, Supplementary Table 1). The variants, along with the parental 2Q1, were recombinantly expressed in bacteria as scFvs (Supplementary

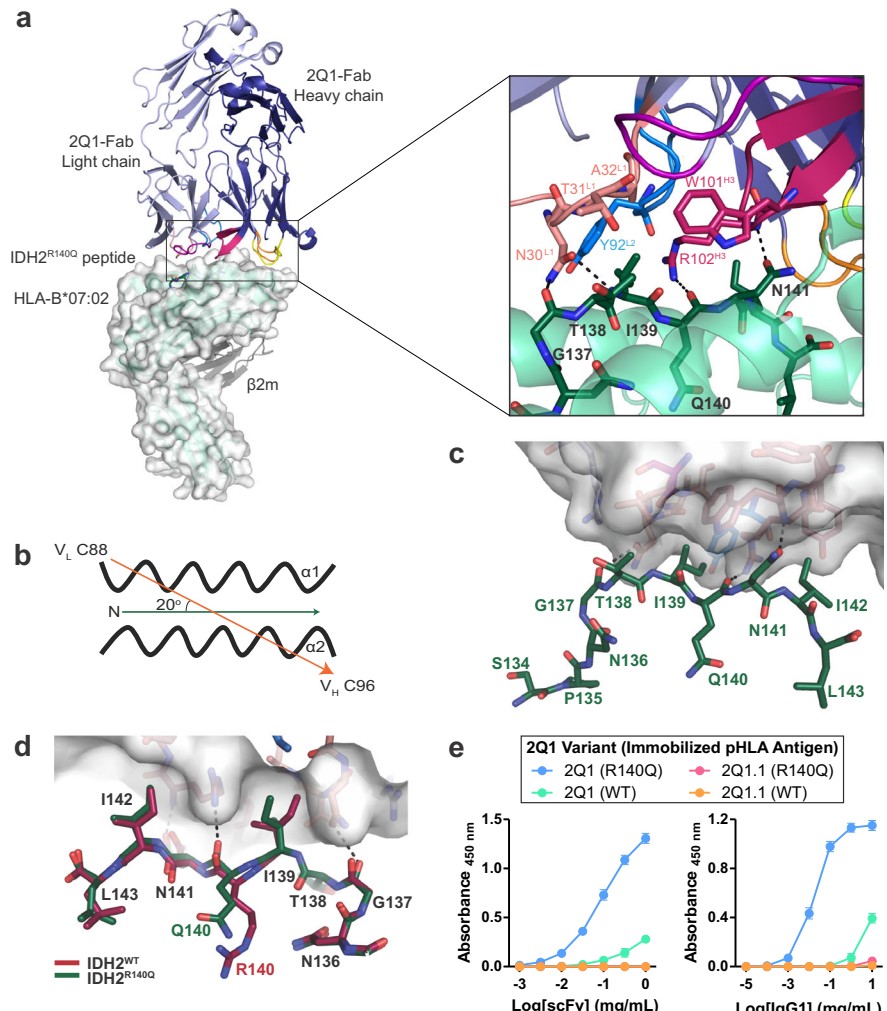

**Fig. 3 Structural determinants of 2Q1 selectivity for IDH2$^{R140Q}$-HLA-B*07:02. a** Overall structure of the 2Q1-Fab bound to IDH2$^{R140Q}$-HLA-B*07:02 (PDB ID 6UJ9). The HLA is shown in green cartoon and white surface. β2-microglobulin is shown as gray cartoon. The 2Q1-Fab light and heavy chains are shown in light and deep blue, respectively, with each of the CDR loops colored as follows: L1—salmon, L2—purple, L3—blue, H1—yellow, H2—orange, H3—raspberry. Detailed interaction of the IDH2$^{R140Q}$ peptide (aa Gly$^{137}$-Asn$^{141}$) with 2Q1-Fab CDR-L1, -L2, and -H3 is shown with a black box. Zoomed view shows interacting residues represented as sticks and hydrogen bonds represented as dashed lines. **b** Schematic representation of the 2Q1-Fab orientation angle. The docking angle was calculated using the web server TCR3d which was based on the C$^{alpha}$ of Cys88 of the disulfide bond of the V$_L$ domain and the C$^{alpha}$ of Cys96 of the disulfide bond of the V$_H$ domain of the 2Q1-Fab. The arrowed line indicates the direction of orientation. **c** Binding of the CDRs to the IDH2$^{R140Q}$-HLA-B*07:02 complex. The 2Q1-Fab is shown in surface representation and the IDH2$^{R140Q}$ peptide as sticks. CDRs are colored as in (**a**). **d** Structural alignment of the IDH2$^{WT}$-HLA-B*07:02 (PDB ID 6UJ8, raspberry) and 2Q1-Fab/IDH2$^{R140Q}$-HLA-B*07:02 (PDB ID 6UJ9, green) structures, showing the interactions between the peptides and 2Q1-Fab. The 2Q1-Fab is shown in white surface with key residues shown as sticks. Hydrogen bonds are depicted as dashed lines. **e** Binding of 2Q1 and 2Q1.1 scFvs or IgG1s to IDH2$^{R140Q}$ or IDH2$^{WT}$ pHLA monomers were assessed by ELISA. Data represent the mean of three technical replicates ± SD. Source data are provided as a Source data file.

Fig. 7b) or were expressed in mammalian cells as IgG1s (Supplementary Fig. 8).

The binding of 2Q1 and its variants was first evaluated by titration ELISA against immobilized IDH2$^{R140Q}$-HLA-B*07:02 and IDH2$^{WT}$-HLA-B*07:02 monomers. Although the majority of the variants tested lost binding to the mutant monomer, the 2Q1.4 scFv and IgG1 (Y103H mutation) retained binding to the mutant monomer with no detectable binding to the WT monomer; this was in contrast to the parental 2Q1 scFv and IgG1, which displayed detectable binding liability to the WT monomer (Fig. 5a, b, Supplementary Fig. 9).

To further probe the improved specificity endowed by the Y103H mutation of 2Q1.4, scanning mutagenesis was employed whereby a peptide library was generated by systematically

substituting each amino acid of the target IDH2$^{R140Q}$ peptide (SPNGTIQNIL) with each of the remaining 19 common amino acids[65]. The 2Q1 and 2Q1.4 IgG1s were then assessed for binding to RPMI-6666 cells loaded with each of the 190 variant peptides via flow cytometry. The binding profile of the 2Q1.4 IgG1 was more specific at multiple positions as compared to the 2Q1 IgG1—including at P3 (IDH2$^{N136}$), P5 (IDH2$^{T138}$), and the position of the mutant amino acid P7 (IDH2$^{R140Q}$) (Fig. 5c, d, Supplementary Fig. 10). The recognition pattern of 2Q1 and 2Q1.4 is also illustrated as sequence logos (Fig. 5c, d). The binding characteristics and specificity of 2Q1.4 were further assessed by SPR. At high concentrations, there was no detectable binding of clone 2Q1.4 to IDH2$^{WT}$-HLA-B*07:02 (Fig. 5e). However, 2Q1.4 showed a fourfold decrease in affinity with a

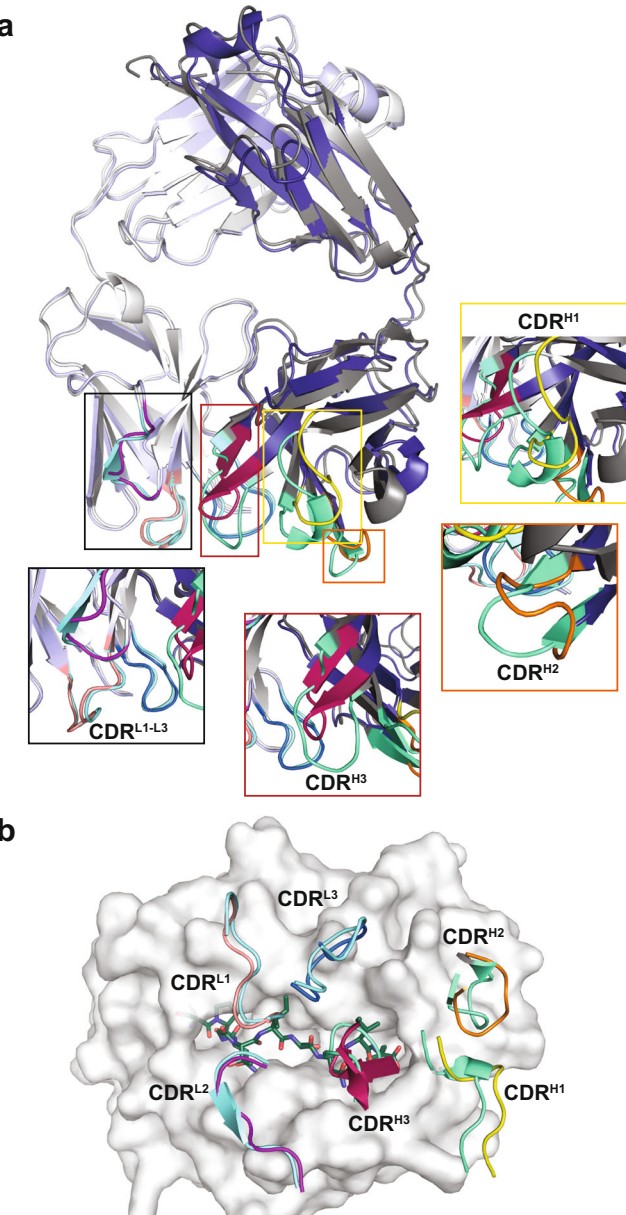

**Fig. 4 2Q1-Fab free form structure. a** Structural alignment of the free 2Q1-Fab (gray) with the 2Q1-Fab bound to the IDH2$^{R140Q}$-HLA-B*07:02 (Light Chain: light blue; Heavy Chain: deep blue). All CDR loops of the free 2Q1-Fab are colored in green. Zoomed views of the CDR loops are highlighted in boxes with CDR-H1 (yellow), -H2 (orange), and -H3 (raspberry) displaying the largest conformation changes. **b** Bird's-eye view of surface representation of the HLA-B*07:02 showing the positions of the 2Q1-Fab CDR loops. The CDRs of the free 2Q1-Fab structure are overlaid with the bound 2Q1-Fab to highlight the conformational change induced upon pHLA binding. The CDRs of the free 2Q1-Fab are shown in light blue (light chain) and pale green (heavy chain). The CDR loops of the bound 2Q1-Fab are colored as in (**a**).

slower off-rate to the mutant monomer compared with the parental 2Q1, with a $K_D = 175$ nM (Fig. 5f, Supplementary Table 2).

To determine whether 2Q1.4 could improve CAR activity against IDH2$^{R140Q}$-bearing target cells, CAR T cells expressing the 2Q1 or 2Q1.4 scFv were co-incubated with RPMI-6666 cells loaded with either IDH2$^{R140Q}$ or IDH2$^{WT}$ peptides. Consistent with the ELISA and peptide positional-scanning results, the

2Q1.4 CAR displayed reduced off-target effects in response to IDH2$^{WT}$ and, surprisingly, improved on-target activity in response to IDH2$^{R140Q}$ as assessed by IFN-γ release (Fig. 6a). Concordantly, the cytotoxicity profiles mirrored those of cytokine release (Fig. 6b).

## Discussion

The paucity of tumor-specific antigens remains a major obstacle for the wider clinical implementation of powerful immunotherapeutic agents such as CAR T cells. Genetic alterations in cancer, particularly hotspot driver gene mutations, offer targeting specificity unmatched by traditional TAA-related targets due to their exclusive presence in tumor but not normal cells. Furthermore, the clonal nature of driver mutations, as well as the essential role they play in cancer progression, provides key advantages over traditional cancer targets that suffer from issues related to tumor heterogeneity and resistance arising from antigen downregulation or loss. Here, we describe a proof-of-concept approach using CAR T cells to exploit such genetic vulnerabilities for the treatment of cancers harboring MANA-derived mutant peptides complexed with HLA.

Various protein- and cellular-engineering methods have been employed to target neoantigens and the like, including TCR-CD3 bispecifics[66], TCRm-CD3 bispecifics[63], TCRm:TCR-T cells[67], TCR-T cells[34], and the TCRm:CAR T-cell approach described herein. While it remains to be determined which class of therapeutic will prove most efficacious in humans, the TCRm:CAR T approach described here allows for the incorporation of antigen-dependent costimulation via CD28 and 4-1BB, a property that has proved essential in the successful translation of CAR T cells to the clinic.

CARs targeting pHLA complexes presenting WT peptides have previously been generated; however, to our best knowledge, there has only been one other report of a CAR targeting a recurrent driver gene mutation[47]. A recently reported mass spectrometry-based assay confirmed the processing and presentation of IDH2$^{R140Q}$-HLA-B*07:02 and quantified the antigen density of IDH2$^{R140Q}$ peptide on transfected COS-7 cells to be ~25 copies per cell[53]. While TCRs, the natural targeting moieties of pHLA complexes, can elicit a response to one or a few antigens, the majority of CARs developed to date appear to require an antigen density threshold on the order of hundreds to thousands per cell[16,61,68–73]. To address this issue, we engineered the CAR design to allow for increased sensitivity with applicability for low antigen density targets such as MANA-derived mutant peptides complexed with HLA as described herein. Optimization of the hinge region of CARs has previously been reported as a means to improve therapeutic activity against cancer cells bearing TAA targets such CD19, PSCA, and ROR1[54–61,74]. However, customization of the hinge region for CARs targeting MANA-pHLA complexes has yet to be explored. Here, we systematically screened four hinges with three neoantigen-specific TCRm scFvs targeting different HLA allele backgrounds (A*02:01, A*03:01, B*07:02) and identified the CD8α hinge as the most sensitive for our given target class of tumor-specific MANA-pHLA complexes. Given that all three target antigens have similar architecture, it is perhaps not surprising that the CD8α hinge universally proved to be the most sensitive—conceivably by providing optimal spacing between the CAR T and target cells in promoting a higher quality immunological synapse. However, it is likely that CAR T cells targeting other antigens would require separate optimization efforts to identify the most efficacious hinge for the given target.

The ternary complex structure described herein provides insights into the unique binding properties of the 2Q1-Fab TCRm to the buried IDH2$^{R140Q}$ mutant residue and the structural basis

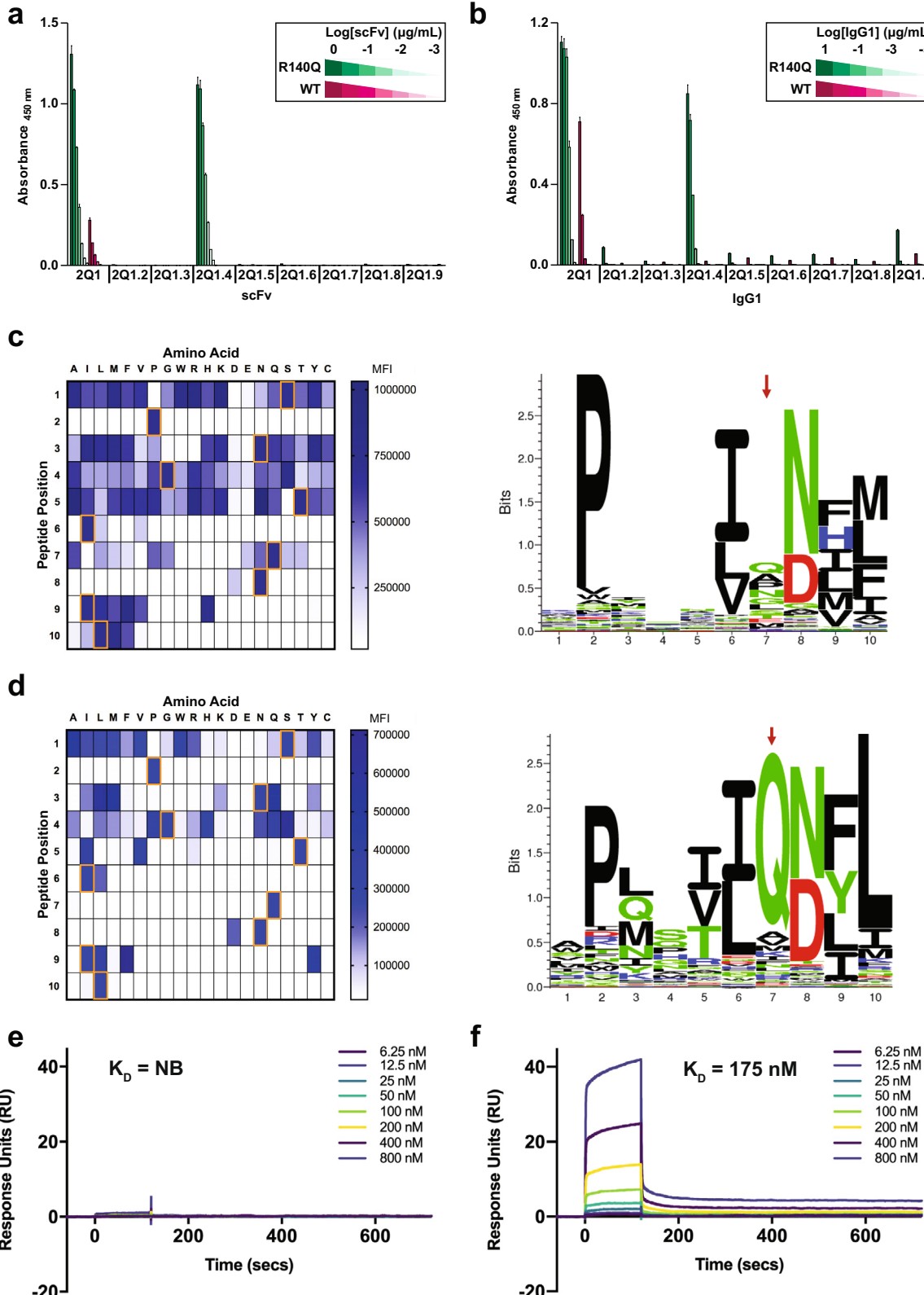

of its specificity. A subtle difference in backbone flexibility and conformation due to the length and interactions of the buried residue allows for the selectivity of the 2Q1-Fab. This is in contrast to known buried viral escape mutants which result in peptide bulging[75]. Importantly, the crystal structure enabled molecular modeling of the TCRm/pHLA interaction which resulted in the identification of a variant (2Q1.4) with improved

binding properties—particularly in specificity for recognition of the mutant pHLA versus the wild-type pHLA—that could then be translated to a CAR targeting platform.

It is not obvious how the simple change of Y103H at CDR-H3 in 2Q1.4 induces such a dramatic improvement in selectivity across the entire peptide sequence. CDR-H3 is important for recognition of the peptide, both directly through making

**Fig. 5 Structure-guided optimization of the 2Q1 clone. a** Binding of 2Q1 and its variant scFvs to IDH2$^{R140Q}$ or IDH2$^{WT}$ pHLA monomers was assessed by ELISA. Data represent the mean of three technical replicates ± SD. Supplementary Fig. 9 includes details about the distribution of the data. **b** Binding of 2Q1 and its variant IgG1s to IDH2$^{R140Q}$ or IDH2$^{WT}$ pHLA monomers was assessed by ELISA. Data represent the mean of three technical replicates ± SD. Supplementary Fig. 9 includes details about the distribution of the data. **c, d** RPMI-6666 cells were loaded with peptide variants from a positional-scanning library and stained with either (**c**) 2Q1 IgG1 or (**d**) 2Q1.4 IgG1. Binding was evaluated by flow cytometry with the gating strategy depicted in Supplementary Fig. 10. Heat maps, on left, were generated from the median fluorescence intensity (MFI). Orange boxes represent the residues in the IDH2$^{R140Q}$ target peptide. Illustrations, on right, of the binding pattern of each respective IgG1 represented as sequence logos. The tallest columns correspond to the positions with the greatest specificity and the red arrows denote the position of the IDH2$^{140}$ residue. **e** 2Q1.4-Fab binding to IDH2$^{WT}$-HLA-B*07:02 was measured by multi-cycle kinetics using SPR with increasing concentrations (6.25, 12.5, 25, 50, 100, 200, 400, 800 nM) of purified 2Q1.4-Fab ($N = 2$). The curves were fit with a 1:1 binding model to calculate the listed $K_D$. NB, no binding. **f** Same as (**e**) using IDH2$^{R140Q}$-HLA-B*07:02. Source data are provided as a Source data file.

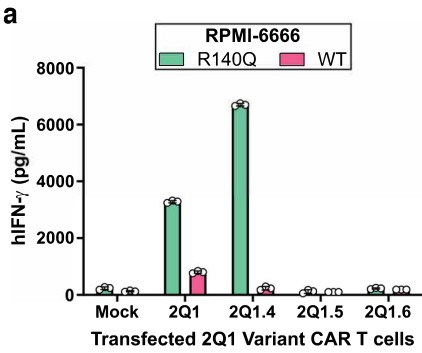
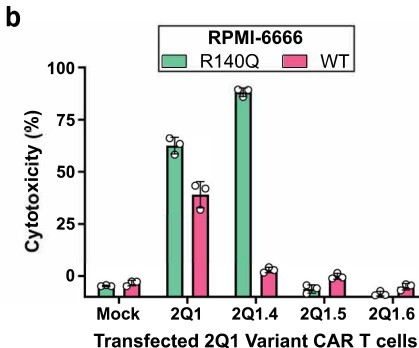

**Fig. 6 Evaluation of 2Q1.4 CAR T-cell activity.** 2Q1 variant CAR T cells were co-incubated with RPMI-6666 cells pulsed with either IDH2$^{R140Q}$ or IDH2$^{WT}$ peptides at an effector-to-target (E:T) ratio of 1:1. **a** T-cell activation, as assessed by IFN-γ release, was measured by ELISA. Bars represent the mean of three technical replicates ± SD, while open circles represent the individual data points. **b** Cytotoxicity was measured by Steady-Glo. Bars represent the mean of three technical replicates ± SD, while open circles represent the individual data points. Source data are provided as a Source data file.

important hydrogen bonds, and indirectly by influencing the conformation of CDR-L1, L2, and L3. We propose that the single amino acid change from tyrosine to histidine propagates conformational changes through CDR-H3 to CDR-L1-L3, thus influencing selectivity across the IDH2 peptide sequence. With regards to mutant versus WT selectivity, the histidine may allow for better positioning of the Arg102 (CDR-H3) to form the key charge-assisted hydrogen bond to the backbone of IDH2$^{R140Q}$.

It is also worthwhile to note that while 2Q1.4 exhibited lower affinity than the parental 2Q1 clone, the 2Q1.4 CAR not only exhibited less off-target activity towards WT-peptide cells, but also increased reactivity towards mutant-peptide presenting cells. This may be explained by the kinetics of the interaction. Although 2Q1.4 has a lower affinity, its off-rate is 112-fold slower, resulting in a longer residency time and better functional potency. The on-rate, however, is three orders of magnitude slower (324-fold). One potential explanation for this is that the conformational ensemble of the parental Fab, 2Q1, is biased towards the bound conformation, and results in a faster association rate. The mutation of heavy chain Y103 to histidine in 2Q1.4 either changes this conformational ensemble in such a way that no longer biases the bound conformation, or the bound conformation is now different. In turn, the more tightly associated pHLA-2Q1.4-Fab results in a slower off-rate. These results would indicate that off-rate is a much better indicator of functional potency than binding affinity alone[76]. Future studies incorporating unbiased high-throughput cell-based screening approaches, in tandem with computational structural modeling, could help in further clarifying the design parameters and improve the hit rate of variant identification.

Other treatment modalities targeting somatic mutations of *IDH2* have been developed. Enasidenib is a small-molecule inhibitor that was recently approved by the FDA for treatment of relapsed or refractory *IDH2*-mutant AML[23,77–79]. Unfortunately,

relapse under enasidenib occurs in the majority of patients, and a recent study has shown that one mechanism of acquired resistance occurs with the emergence of secondary-site *IDH2* mutations (Q316E and I319M) that, together with mutant R140Q, disrupt the binding of enasidenib and induce 2HG production[28,80,81]. This resistance mechanism is similar to secondary-site resistance mutations that arise following treatment with other tyrosine kinase inhibitors[82]. In comparison, our strategy to target IDH2$^{R140Q}$ via its surface presentation in complex with HLA not only provides an alternative method to target this critical mutation, but also circumvents secondary-site resistance mutations common with small-molecule inhibitors. This approach could also be extended to treat other cancers harboring an IDH2 mutation where targeted therapies have not yet been successful.

One limitation of our study stems from the lack of testing in cells that endogenously express IDH2$^{R140Q}$ and HLA-B*07:02. While this would be the most desirable setting to assay our CAR T cells, to our best knowledge, we were unable to find any commercially-available cell line that endogenously harbors the requisite IDH2 mutation and the corresponding HLA-restriction element.

Collectively, we have shown that it is possible to develop a CAR T-cell-based therapeutic modality incorporating an engineered hinge domain and a structure-guided binding moiety that can target a hotspot driver gene mutation presented in the context of a common HLA allele. This approach builds on the clinical success of CAR T cells in treating hematological malignancies and serves as a proof-of-concept study for extending such a therapeutic platform to target other cancers with mutations that are challenging to treat by conventional means. Although subsequent studies including those involving preclinical animal models and human clinical trials will ultimately determine whether such an approach can successfully treat cancer patients, the present work described herein lays the foundation for such future exploration.

## Methods

**Cell lines.** All cell lines (RPMI-6666, COS-7, T2), except for T2A3 cells (a kind gift from Eric Lutz and Elizabeth Jaffee, JHU) were procured from ATCC (Manassas, VA). RPMI-6666 cell lines were maintained in RPMI-1640 media (ATCC, Manassas, VA) supplemented with 20% fetal bovine serum (FBS) (HyClone Defined, GE Healthcare, Marlborough, MA) and 1% Penicillin-Streptomycin (Thermo Fisher Scientific, Waltham, MA). COS-7 cells were maintained in Dulbecco's modified Eagle medium (DMEM) (high glucose, pyruvate) media (Thermo Fisher Scientific, Waltham, MA) supplemented with 10% FBS (HyClone Defined, GE Healthcare, Marlborough, MA) and 1% Penicillin-Streptomycin (Thermo Fisher Scientific, Waltham, MA). T2 and T2A3 cell lines were maintained in RPMI-1640 media (ATCC, Manassas, VA) supplemented with 10% FBS (HyClone Defined, GE Healthcare, Marlborough, MA) and 1% Penicillin-Streptomycin (Thermo Fisher Scientific, Waltham, MA). All cells were maintained at 37 °C under 5% $CO_2$.

**Peptides and pHLA monomer complexes.** Peptides were synthesized at a purity of >90% by Peptide 2.0 (Chantilly, VA), resuspended in dimethyl sulfoxide (DMSO) at 10 mg/mL, and stored at −20 °C. To produce pHLA monomer complexes for phage display, ELISA, and SPR, HLA molecules were expressed in *Escherichia coli* (*E. coli*), refolded with peptide and β2-microglobulin, purified using size exclusion, and biotinylated (FHCRC Immune Monitoring Lab, Seattle, WA; Baylor MHC Tetramer Core, Houston, TX). pHLA monomer complexes were confirmed to be folded by ELISA via detection with α-HLA class I (clone W6/32, 1:1000 dilution) antibody (BioLegend, San Diego, CA) prior to use[83].

pHLA-B*07:02 complexes used for crystallography were prepared as follows. Plasmids for HLA-B*07:02 and β2-microglobulin were received from the NIH Tetramer Facility (Atlanta, GA), separately transformed into BL21(DE3) cells, and expressed in inclusion bodies using autoinduction medium[84–86]. The refolding was performed as described previously[39,40]. Briefly, HLA-B*07:02 and β2-microglobulin inclusion bodies were combined in refolding buffer containing 100 mM Tris pH 8.3, 400 mM L-arginine, 2 mM ethylenediaminetetraacetic acid (EDTA), 5 mM reduced glutathione, 0.5 mM oxidized glutathione, 2 mM phenylmethylsulfonyl fluoride (PMSF), and 30 mg/L of either the WT (amino acids 134–143, SPNGTIRNIL) or mutant (amino acids 134–143, SPNGTIQNIL) peptide first dissolved to 30 mg/mL in DMSO and added to the refolding buffer. The resultant mixture was stirred at 4 °C for 2 days, with two further additions of HLA-B*07:02 inclusion bodies on day 2, concentrated to 10 mL and purified by size-exclusion chromatography on a Superdex 75 26/60 column (GE Healthcare, Marlborough, MA). Purified pHLA-B7 was concentrated to 12–13 mg/mL and stored at −80 °C until use.

**Phage display.** Phage clones binding to IDH2^R140Q-HLA-B*07:02 were isolated as previously described[39,40]. In brief, two scFv phage display libraries—one built with degenerate codon (NNK) technology[39] (1st generation, estimated complexity of $5.5 \times 10^9$) and the other built with TRIM technology[40] (2nd generation, estimated complexity of $3.6 \times 10^{10}$)—were screened for binders that could selectively bind IDH2^R140Q-HLA-B*07:02 monomer but not IDH2^WT-HLA-B*07:02 monomer. Multiple rounds of selection included iterations of negative selection with the IDH2^WT monomer followed by positive selection with the IDH2^R140Q mutant monomer. Each panning campaign employed competitive phase panning[39], whereby the ratio of mutant-to-WT monomer was progressively skewed towards the WT monomer to enforce selection of phage specific to the mutant monomer.

**ELISA.** For all pHLA monomer-based ELISAs, biotinylated monomers were coated onto EvenCoat Streptavidin Coated Plates (R&D Systems, Minneapolis, MN) at 1 μg/mL and allowed to bind overnight. For phage ELISAs, precipitated phage or phage-laden supernatant were diluted to the indicated dilutions, allowed to bind to the monomer-coated plate, and detected by sequential application of α-fd/M13 bacteriophage (Novus Biologicals, Centennial, CO) and α-Rabbit IgG HRP antibodies (Thermo Fisher Scientific, Waltham, MA) at 1:3000 and 1:10,000 dilutions, respectively. For scFv ELISAs, recombinant scFvs were diluted to the indicated concentrations, allowed to bind to the monomer-coated plate, and detected by α-FLAG [M2] HRP antibody (Abcam, Cambridge, UK) at a 1:5000 dilution. For IgG ELISAs, mammalian-expressed IgGs were diluted to the indicated concentrations, allowed to bind the monomer-coated plate, and detected by α-human IgG HRP antibody (Thermo Fisher Scientific, Waltham, MA) at a 1:5000 dilution. All ELISAs were developed by addition of 3,3′,5,5′-tetramethylbenzidine (TMB) substrate (BioLegend, San Diego, CA) and stopped with an equal volume of 1N Sulfuric Acid Solution (Thermo Fisher Scientific, Waltham, MA).

**Flow cytometry.** PE-conjugated α-Rabbit IgG (1 μL/sample) and APC-conjugated α-human IgG Fc (clone HP6017, 5 μL/sample) were obtained from BioLegend (San Diego, CA), and α-fd/M13 bacteriophage (1 μL/sample) was obtained from Novus Biologicals (Centennial, CO). Flow cytometry was performed either on an LSRII cytometer (BD Biosciences, San Jose, CA) or an IntelliCyt iQue Screener PLUS (Sartorius, Gottingen, Germany). Analysis was performed either with FACSDiva or FlowJo software (BD, Franklin Lakes, NJ).

**mRNA generation.** Human codon-optimized constructs were synthesized (GeneArt, Thermo Fisher Scientific, Waltham, MA) and cloned into the mammalian expression vector pCI (Promega, Madison, WI). Per manufacturers' instructions, the T7 mScript Standard mRNA Production System Kit (CELLSCRIPT, Madison, WI) was used to synthesize Cap 1 mRNA, followed by purification using the MEGAclear Transcription Clean-Up Kit (Thermo Fisher Scientific, Waltham, MA). Prior to electroporation, RNA TapeStation (Agilent, Santa Clara, CA) analysis was performed to determine mRNA purity, integrity, and transcript size.

**T-cell activation, culture, and electroporation.** mRNA-modified T cells were generated as follows. Human cellular products were procured from commercial sources; in brief, Ficoll-Paque PLUS (GE Healthcare, Marlborough, MA) gradient centrifugation of whole blood from healthy volunteer donors was used to isolate peripheral blood mononuclear cells (PBMCs), followed by isolation of CD3 T cells via negative selection (STEMCELL Technologies, Vancouver, Canada; Cellero, Seattle, WA). Activation and expansion of primary human CD3 T cells were performed using Dynabeads Human T-Activator CD3/CD28 (Thermo Fisher Scientific, Waltham, MA). Unless otherwise specified, primary T cells were maintained in RPMI-1640 (ATCC, Manassas, VA) supplemented with 10% FBS (HyClone Defined, GE Healthcare, Marlborough, MA), 1% Penicillin-Streptomycin (Thermo Fisher Scientific, Waltham, MA), and 100 IU/mL recombinant human interleukin-2 (Proleukin, Prometheus Laboratories, San Diego, CA). Cells were maintained at 37 °C under 5% $CO_2$. Electroporation of the primary human CD3 T cells was performed using the BTX ECM 2001 Electro Cell Manipulator (Harvard Apparatus, Holliston, MA) in a 0.2 cm cuvette (Bio-Rad, Hercules, CA). Briefly, 125 μL of electroporation volume containing $2 \times 10^7$ cells/mL with the specified amount of mRNA and diluted in Opti-MEM (Thermo Fisher Scientific, Waltham, MA) was electroporated with a square wave pulse of 200 V and 16 ms. T-cell viability following electroporation was assessed by propidium iodide (PI) dye exclusion (Thermo Fisher Scientific, Waltham, MA).

**Target cell line generation.** To generate exogenous target cell lines, 96-well plates were seeded with COS-7 cells, and Lipofectamine 3000 (Thermo Fisher Scientific, Waltham, MA) was used to transfect the cells with indicated combinations of plasmids in the pcDNA3.1 backbone (Thermo Fisher Scientific, Waltham, MA). Cells were washed 24 h after transfection prior to use as target cells for T-cell co-culture assays. To generate a luciferase-expressing RPMI-6666 cell line, cells were transduced with RediFect Red-Fluc-GFP lentiviral particles (PerkinElmer, Waltham, MA). Transduced cells were sorted for GFP expression with a FACSAria Fusion (BD Biosciences, San Jose, CA) to obtain a purely transduced population. Engineered target cell lines were authenticated by short-tandem repeat (STR) profiling, and parental origin (100% exact match of 8 core STR loci) was confirmed by ATCC Cell Line Authentication Service (Manassas, VA).

**Peptide pulsing.** For peptide pulsing, cells were washed with serum-free media prior to incubation with the indicated concentrations of peptide overnight at a concentration of $1 \times 10^6$ cells/mL with 10 μg/mL β2-microglobulin (β2M) (ProSpec, East Brunswick, NJ) in serum-free media at 37 °C.

**In vitro T-cell assays.** Engineered T cells and target cell lines were combined at the specified effector-to-target (E:T) ratios in flat-bottom 96-well plates, followed by a 4-h or overnight incubation. Clarified cell culture supernatant was quantified for cytokine levels using Quantikine ELISA kits (R&D Systems, Minneapolis, MN) per manufacturer's instructions. Viability of the target cell lines was measured by Steady-Glo Luciferase Assay System for RPMI-6666 cells, per manufacturer's instructions (Promega, Madison, WI).

**Peptide-scanning mutagenesis.** Peptides for the positional-scanning library were synthesized as crude peptides (ELIM Biopharm, Hayward, CA). The peptide library was resuspended in dimethylformamide (DMF) at 10 mg/mL and stored at −20 °C. RPMI-6666 cells were pulsed with 50 μg/mL peptide as described above, stained with 2 μg of either 2Q1 or 2Q1.4 IgG1, followed by fluorophore-conjugated α-human IgG Fc and analyzed via flow cytometry. The sequence logos were generated by first transforming the median fluorescence intensity (MFI) into a "pseudo-frequency", in which we first subtracted the background MFI value of the no peptide condition and divided the MFI of a particular variant peptide by the sum of the MFIs of all the 20 peptides with amino acid variation at the same position. The "pseudo-frequency" was then submitted to Seq2Logo 2.0 using the Shannon algorithm[87] to generate the illustrations shown.

**scFv and IgG expression and purification.** For scFv protein production, DNA fragments encoding scFvs were cloned into the pAP-III$_6$ vector containing C-terminal FLAG- and HIS-tags and transformed into *E. coli*[88]. Proteins were subsequently purified using nickel chromatography (AxioMx, Abcam, Cambridge, UK). For IgG protein production, the light and heavy chain variable region sequences of scFvs were grafted onto the respective constant chains of trastuzumab and cloned into the pcDNA3.4 backbone with a mouse IgKVIII leader peptide (Thermo Fisher Scientific, Waltham, MA). Light and heavy chain plasmids were

co-transfected at a 1:1 ratio into Expi293 cells, and the supernatant was harvested 6 days following transfection. Purification was conducted with a HiTrap Protein A HP column, and the resultant eluate was dialyzed against PBS and stored at −80 °C (GeneArt, Thermo Fisher Scientific, Regensburg, Germany).

**Fab preparation.** For the generation of 2Q1-Fab fragments, ~8 mg of full-length 2Q1-IgG1 antibody was mixed with 0.5 mL of a 50% Immobilized Papain slurry (Thermo Fisher Scientific, Waltham, MA) pre-activated with digestion buffer (20 mM sodium phosphate buffer, pH 7.0, 10 mM EDTA) containing 20 mM cysteine-HCl. The mixture was incubated at 37 °C overnight with constant shaking at 200 rpm. The digested 2Q1 antibody was separated from the immobilized resin by a gravity resin separator and washed with 10 mM Tris-HCl, pH 7.5. Newly generated 2Q1-Fab fragments were further purified by cation-exchange chromatography using a Mono S column (GE Healthcare, Marlborough, MA) and eluted using a linear gradient of 0–500 mM NaCl in 50 mM HEPES pH 7 buffer. A similar procedure was performed for the generation of the 2Q1.4-Fab.

The 2Q1-Fab fragments were concentrated, mixed with IDH2$^{R140Q}$-HLA-B*07:02 at a 1:2 molar ratio, and incubated on ice for 30 min. The 2Q1-Fab/IDH2$^{R140Q}$-HLA-B*07:02 mixture was evaluated by size-exclusion chromatography on a Superdex 200 Increase 10/300 column (GE Healthcare, Marlborough, MA). The fractions of ~98% pure pHLA-B*07:02/2Q1-Fab complex were pooled, concentrated to 10.8 mg/mL, and stored at −80 °C.

**Crystallization, data collection, and structure determination.** Crystals of IDH2$^{WT}$-HLA-B*07:02 were prepared using hanging-drop vapor diffusion with a reservoir solution of 0.1 M MES pH 6.5, 25% (w/v) PEG 4000. Crystals of IDH2$^{R140Q}$-HLA-B*07:02 were prepared by vapor diffusion in hanging drops with a reservoir solution of 0.1 M MES pH 6.5, 0.2 M NaCl, 20–35% (w/v) PEG 4000. Crystals of 2Q1-Fab were prepared with a reservoir solution of 0.1 M Bis-Tris pH 5.5, 25% (w/v) PEG 3350, 0.2 M ammonium sulfate. Crystals of IDH2$^{R140Q}$-HLA-B*07:02 in complex with 2Q1-Fab were prepared with a reservoir solution of 0.1 M CHES pH 9.5, 20% (w/v) PEG 8000. All crystals were cryoprotected in their own mother liquor and flash-cooled. Data were collected at National Synchrotron Light Source-II at beamlines 17-ID-1 (IDH2$^{R140Q}$-HLA-B*07:02, 2Q1-Fab/IDH2$^{R140Q}$-HLA-B*07:02 complex, 2Q1-Fab) on a DECTRIS Eiger X 9M detector and 17-ID-2 (IDH2$^{WT}$-HLA-B*07:02) on a DECTRIS Eiger X 16M detector. Data of the 2Q1-Fab/IDH2$^{R140Q}$-HLA-B*07:02 were collected with a beam size of 7 × 5 over a vector of ~30 μm for 180° degrees with an oscillation width of 0.5 and an exposure of 0.03 s per frame[89] (Supplementary Fig. 4c, d). Data of the 2Q1-Fab were collected for 180° with an oscillation width of 0.2 and an exposure of 0.03 s per frame. Datasets were indexed, integrated, and scaled using XDS[90]. The structure for IDH2$^{R140Q}$-HLA-B*07:02 was determined by molecular replacement with MOLREP[91] using PDB entry 3VCL[92] as the search model. The structure for IDH2$^{WT}$-HLA-B*07:02 was determined by molecular replacement with MOLREP[91] using IDH2$^{R140Q}$-HLA-B*07:02 as the search model. The structure for the 2Q1-Fab/IDH2$^{R140Q}$-HLA-B*07:02 complex was determined by molecular replacement with PHASER[93] using IDH2$^{R140Q}$-HLA-B*07:02 and PDB entry 6DF0[94] as search models. The structure for 2Q1-Fab alone was determined by molecular replacement with MOLREP[91] using the 2Q1-Fab from the complex structure as a search model. All datasets were refined using iterative rounds of refinement with REFMAC5[95,96] and manual rebuilding in Coot[97]. Structures were validated using Coot and the PDB deposition tools. Refinement statistics are summarized in Table 1. Contact residues were identified with the CONTACT program in the CCP4 suite and were defined as residues containing an atom 4 Å or less from a residue in the pair[95]. Figures were rendered in PyMOL (version 2.2.3, Schrödinger, New York, NY).

**Calculation of docking and incident angle.** The calculations were performed using the web server TCR3d (https://tcr3d.ibbr.umd.edu/) where the $x$–$y$ plane is parallel to the peptide and HLA helices α1 and α2[98,99].

**Modeling.** Modeling was performed using the Biologics Suite, accessed using the Maestro interface (version 2019, Schrödinger, New York, NY). Programs used for modeling were accessed through SBGrid[100]. The 2Q1-Fab/IDH2$^{R140Q}$-HLA-B*07:02 complex structure was prepared using the Protein Preparation Wizard[101]. The amino acid positions on 2Q1 that bound within 8 Å of the mutant glutamine residue in the HLA-bound peptide were mutated in silico to each of the nineteen other naturally occurring amino acids, and the change in affinity was calculated[102]. Overall, nine mutations were selected for evaluation in vitro. Eight of them were chosen based on their potential predicted enhanced binding affinity while the ninth (2Q1.1, R102A) was selected to test the relevance of the hydrogen bond R102-IDH2$^{R140Q}$.

**Surface plasmon resonance.** All SPR experiments were performed on a Biacore T200 (GE Healthcare, Marlborough, MA) optical biosensor at 25 °C in duplicate. Approximately 215–240 response units of biotinylated IDH2$^{R140Q}$ mutant and IDH2$^{WT}$ pHLA-B7 were loaded onto flow cells (Fc) 2 and 3, respectively, of a streptavidin sensor chip. Multi-cycle kinetics were performed with increasing concentrations (6.25, 12.5, 25, 50, 100, 200, 400, 800 nM) of purified 2Q1- and

2Q1.4-Fab flowed over Fc 1–4. Binding responses for kinetic analysis were both reference-subtracted and blank-subtracted. The curves were fit with a 1:1 binding model using Biacore Insight Evaluation Software (GE Healthcare, Marlborough, MA).

**Reporting summary.** Further information on research design is available in the Nature Research Reporting Summary linked to this article.

## Data availability
The final coordinates of IDH2$^{R140Q}$-HLA-B*07:02, IDH2$^{WT}$-HLA-B*07:02, 2Q1-Fab/IDH2$^{R140Q}$-HLA-B*07:02, and 2Q1-Fab have been deposited in the PDB with accession codes 6UJ7, 6UJ8, 6UJ9, and 7KGU, respectively. Source data are provided with this paper.

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

## Acknowledgements

This work was supported by The Virginia and D.K. Ludwig Fund for Cancer Research; Lustgarten Foundation for Pancreatic Cancer Research; The Commonwealth Fund; The Burroughs Wellcome Career Award For Medical Scientists; The Bloomberg~Kimmel Institute for Cancer Immunotherapy; National Institutes of Health Cancer Center Support Grant P30 CA006973; National Cancer Institute Grant R37 CA230400. S.P. was supported by the National Institutes of Health T32 Grant 5T32CA009071-38, and the SITC-Amgen Cancer Immunotherapy in Hematologic Malignancies Fellowship. M.F.K. was supported by the National Institutes of Health T32 Grant AR048522. J.D., B.J.M., A.H.P., and S.R.D. were supported by NIH T32 Grant GM136577. The authors would like to thank the NIH Tetramer Core Facility (Emory University, Atlanta, CA) for providing the HLA-B*07:02 and β2-microglobulin plasmids. Work at the AMX (17-ID-1) and FMX (17-ID-2) beamlines is supported by the National Institutes of Health, National Institute of General Medical Sciences (P41GM111244), and by the DOE Office of Biological and Environmental Research (KP1605010), and the National Synchrotron Light Source-II at Brookhaven National Laboratory is supported by the DOE Office of Basic Energy Sciences under contract number DE-SC0012704 (KC0401040). The authors acknowledge use of the Eukaryotic Tissue Culture Facility at JHU and assistance from the manager, Dr. Yana Li. The authors acknowledge the use of the Biacore Molecular Interaction Shared Resource (BMISR) at Georgetown University, which is supported by the National Institutes of Health (P30CA51008). The authors would also like to thank Dr. Brian Pierce and Dr. Ragul Gowthaman for tailoring the web server TCR3d for antibodies.

## Author contributions

K.M.W., P.T., J.D., and E.H.H. contributed equally. Conceptualization: M.S.H., M.S.M., J.D., B.V., S.Z., and S.B.G.; methodology: M.S.H., M.S.M., P.T., J.D., K.M.W., E.H.H., and M.B.M.; investigation: M.S.H., M.S.M., P.T., J.D., K.M.W., E.H.H., B.J.M., T.Y.A., and M.B.M.; analysis and interpretation of data: M.S.H., M.S.M., P.T., J.D., K.M.W., E.H.H., B.J.M., P.A.A., A.H.P., S.P., S.R.D., M.F.K., C.B., D.M.P., N.P., K.W.K., B.V., S.Z., and S.B.G.; writing—original draft: M.S.H., M.S.M., & K.M.W.; writing—review & editing: M.S.H., M.S.M, P.T., J.D., K.M.W., E.H.H., B.J.M., T.Y.A., M.B.M., P.A.A., A.D.S., A.H.P, S.P., S.R.D., M.F.K., C.B., D.M.P., N.P., K.W.K., B.V., S.Z., and S.B.G.; supervision: C.B., N.P., K.W.K., B.V., S.Z., and S.B.G.

## Competing interests

B.V., K.W.K., and N.P. are founders of Thrive Earlier Detection. K.W.K. and N.P. are consultants to and were on the Board of Directors of Thrive Earlier Detection. B.V., K.W.K., N.P., and S.Z. own equity in Exact Sciences. B.V., K.W.K., N.P., S.Z., and D.M.P. are founders of and serve or may serve as consultants to ManaT Bio, and hold or may hold equity in ManaT Holdings, LLC. B.V., K.W.K., N.P., and S.Z. are founders of, hold equity in, and serve as consultants to Personal Genome Diagnostics. S.Z. has a research agreement with BioMed Valley Discoveries. S.B.G. is a founder and holds equity in AMS. K.W.K. and B.V. are consultants to Sysmex, Eisai, and CAGE Pharma and hold equity in CAGE Pharma. B.V. is also a consultant to Catalio. K.W.K., B.V., S.Z., and N.P. are consultants to and hold equity in NeoPhore. N.P. is an advisor to and holds equity in CAGE Pharma. C.B. is a consultant to Depuy-Synthes and Bionaut Pharmaceuticals. The companies named above, as well as other companies, have licensed previously described technologies related to the work described in this paper from Johns Hopkins University. B.V., K.W.K., S.Z., N.P., and C.B. are inventors on some of these technologies. Licenses to these technologies are or will be associated with equity or royalty payments to the inventors as well as to Johns Hopkins University. The terms of all these arrangements are being managed by Johns Hopkins University in accordance with its conflict of interest policies. M.F.K. received personal fees from Bristol Myers Squibb and Celltrion. D.M.P. reports grant and patent royalties through institution from Bristol Myers Squibb, grant from Compugen, stock from Trieza Therapeutics and Dracen Pharmaceuticals, and founder equity from Potenza; being a consultant for Aduro Biotech, Amgen, AstraZeneca (MedImmune/Amplimmune), Bayer, DNAtrix, Dynavax Technologies Corporation, Ervaxx, FLX Bio, Rock Springs Capital, Janssen, Merck, Tizona, and Immunomic Therapeutics; being on the scientific advisory board of Five Prime Therapeutics, Camden Nexus II, WindMIL; and being on the board of directors for Dracen Pharmaceuticals. The remaining authors declare no competing interests.
