## [Peer Review File · Nature Communications]

REVIEWER COMMENTS

Reviewer #1 (Remarks to the Author):

Hwang et al performed structural analysis of the complex of TCRm/CAR binding moieties and HLA neoantigen, specifically, tumor- specific isocitrate dehydrogenase 2 (IDH2) mutation. CAR-T design targeting HLA/neoantigen represents a new frontier in cancer immunotherapy, however, their development has been hindered due to the lack of design principle that can inform such CAR designs. Investigating structural and compositional variables is therefore important and can provide insights to guide future CAR-T design. Using IDH2 mutant as a model target, the study is also carefully and systemically conducted especially with respect to structural analysis of the pHLA and binding moiety complex. The authors further demonstrated they can use structure analysis to guide and optimize the functions and performance of targeting moieties. I have several comments below to address:

- (1) While it is understandable the binding moieties (scFv, Fab) (not CAR-T cells) were used to probe structural insights with pHLA and it is comforting to see such data can be replicated when these moieties are used in CAR-T cell itself (Figure 6), it should be acknowledged that the function of “binder” does not always translate to the ultimate function of CAR-T cells. That said, would not it be more efficient for translation using an approach where CAR-T library is constructed and their function screened first to identify optimal CARs with their binding motif structure then analyzed? A discussion on this subject will be useful.
- (2) Along that line, the broad utility of the structural insights learned from this study to guide CAR-T design for different neoantigen targets is unclear. Though it would be too much to ask to demonstrate a second case, it'd be great if the authors can more clearly define the structural design principles or considerations that other researchers should use when design future neoantigen CARs.
- (3) It'd also be useful to elaborate the advantages of CAR-T for neoantigen targeting compared to TCR-engineered T cells or TCRm. With this structural insights and optimized binding moieties, why not just engineer and use TCR-T cells for treatment which after all are natural pMHC.

Reviewer #2 (Remarks to the Author):

Hwang et al report a very nice structure-function study on a clinically impactful problem. The authors have been very thorough in their analyses, with the crystal structures of pHLA binary complexes, the Fab unliganded, and the Fab ternary complex. The structures – which are documented well, are also supported by biophysical measurements. While the basis for the selectivity for the mutant epitope is somewhat obscure, the authors have provided some plausible explanations as to the mechanism. Often buried MHC polymorphisms, or HLA-buried viral escape mutants, can unexpectedly impact on TCR recognition, so perhaps the authors can reflect on this. I note that the mutant results in a slower off rate (Fig. 1D vs 1E), but the authors do not speculate why. Did authors look at B-factors of the binary complexes? Examining thermal stability of the pHLA binary complexes could be informative.

It's a comprehensive, well-written and nicely presented paper that requires limited modifications. Minor comments:

SPR experiments. N=? independent measurements

As the ternary complex was solved at modest resolution, the structural differences observed could just be due to coordinate error. Authors should acknowledge this
ref 48 is incorrect wrt TCR-pMHC structures.

Any comments/thoughts why the Fab is specific for the B*07:02 allomorph vs other HLA molecules?

Reviewer #3 (Remarks to the Author):

This interesting study reports the development of pHLA-targeting, TCR-mimicking CARs that recognize the IDH2 R140Q mutation presented by HLA-B*0702. Phage display with positive selection with the mutant and negative selection with the wildtype pHLA was used to generate several scFV clones, of which one ('2Q1') was characterized in detail biophysically and structurally. Surprisingly, the mutant and wildtype pHLA complex structures are highly similar, with the mutated position facing inward toward the MHC and only very minor backbone differences between the two peptides. Nonetheless, the 2Q1 scFV is able to discriminate between the two with reasonable specificity (but still some binding to the WT pHLA), in part through a side chain-backbone hydrogen bond from R102(H3) to the carbonyl oxygen of 140Q. Structural modeling is then used to identify a mutation in 2Q1 (Y103H) that improves the specificity for the mutant by weakening binding to the wild type pHLA (and perhaps also to the mutant, but not by as much). Overall, there are a number of interesting findings here and the study seems well worth publishing. The observation of such strong specificity at an inward-facing peptide position is an important finding, which together with the Y103H mutation highlights once again the subtlety of macromolecular interactions.

I have a few minor questions/suggestions, as follows.

When I compare the mutant and wildtype peptide backbones in Figure 2F it does look like the mutant sits a little higher and perhaps has the carbonyl oxygen angled slightly differently. But it's difficult to tell from the angle. Could the wildtype peptide from the pMHC structure be superimposed onto the mutant peptide in the 2Q1 co-complex structure, to see if the R102 Hbond could still be made? This would not be at all conclusive but I'm curious whether it would provide any insight. Of course there is ample room for structural relaxation here.

The description of the structural modeling is quite brief: how were the mutations selected? On the basis of predicted binding affinities? More details would be useful here, even if the answer is that it was mainly human intuition in the end.

The sequence logos in Fig 5C,D do not look right at all. Traditionally, the height of a letter stack in this type of visualization is related to the degree of conservation at that position. So for example logos of allele-specific, Class I-presented peptides typically show tall columns at position 2 and N, the anchor positions. I don't know how the flow data was converted into sequence logos but it does not seem to match one's expectations. For example, the position 2 stack is very small, even though the heat map to the left suggests high specificity. Similarly at position 7, the heat map seems to suggest that 2Q1.4 is more specific. But the letter stack is higher for 2Q1. Please first of all clarify how the heatmaps are turned into logos, and consider using a more intuitive algorithm so that stack height is indicative of specificity.

REVIEWER COMMENTS

Reviewer #1 (Remarks to the Author):

Hwang et al performed structural analysis of the complex of TCRm/CAR binding moieties and HLA neoantigen, specifically, tumor- specific isocitrate dehydrogenase 2 (IDH2) mutation. CAR-T design targeting HLA/neoantigen represents a new frontier in cancer immunotherapy, however, their development has been hindered due to the lack of design principle that can inform such CAR designs. Investigating structural and compositional variables is therefore important and can provide insights to guide future CAR-T design. Using IDH2 mutant as a model target, the study is also carefully and systemically conducted especially with respect to structural analysis of the pHLA and binding moiety complex. The authors further demonstrated they can use structure analysis to guide and optimize the functions and performance of targeting moieties. I have several comments below to address:

(1) While it is understandable the binding moieties (scFv, Fab) (not CAR-T cells) were used to probe structural insights with pHLA and it is comforting to see such data can be replicated when

these moieties are used in CAR-T cell itself (Figure 6), it should be acknowledged that the function of “binder” does not always translate to the ultimate function of CAR-T cells. That said, would not it be more efficient for translation using an approach where CAR-T library is constructed and their function screened first to identify optimal CARs with their binding motif structure then analyzed? A discussion on this subject will be useful.

We thank this Reviewer for bringing this to our attention and for the suggestion. We certainly agree that soluble “surrogate” antibody fragments, including those of scFv and Fab, can behave differently than their surface-bound, cellularly-presented counterparts. While we were fortunate that the structural insights gleaned from the Fab ternary complex was able to translate to CAR T cell functional assays, we fully acknowledge that this may not always be the case. The suggestion by this Reviewer to generate a CAR T library for screening is certainly an intriguing thought and one that has some precedence in the field (PMID: 32827460). From our past experiences generating neoantigen-specific TCRm antibodies (PMID: 26216968, 31690625, 33649166, 33649101), we have found the most success with phage-display technology in isolating antibodies with high specificity as the phage panning protocol can be highly customized in favor of stringent negative selection. We have also found that isolating neoantigen-specific TCRm antibodies benefits from an extremely diverse starting “naïve” library, something that is much more feasible with phage-display technology versus primary T cell knock-in library strategy approaches. Furthermore, phage-display technology has allowed our group to graft isolated scFvs to other therapeutic formats outside of the CAR T realm (e.g. bispecific antibodies; PMID: 33649166, 33649101). However, a CAR T library screen via a “cell-based panning” approach certainly has significant advantages in directly optimizing the most downstream element and this Reviewer’s suggestion is well-taken.

We have modified the Results section to now include the following paragraph addendum:

“A phage display approach was employed to not only allow for facile grafting to multiple therapeutic platforms, but also to enable protocol customization in favor of stringent negative selection.”

(2) Along that line, the broad utility of the structural insights learned from this study to guide CAR-T design for different neoantigen targets is unclear. Though it would be too much to ask to demonstrate a second case, it’d be great if the authors can more clearly define the structural design principles or considerations that other researchers should use when design future neoantigen CARs.

We thank this Reviewer for highlighting an area to clarify and expand upon. From structural studies from ourselves and others on neoantigen pHLA complexes (PMID: 33649166, 32518267, 32461371), this particular IDH2^{R140Q} target is certainly unique with respect to its buried mutant residue. One intriguing aspect of our study is the identification of a “universal” hinge element that imparts improved CAR T sensitivity to pHLA neoantigen targets (Fig. 1A/B). As pHLA neoantigen targets typically exist at extremely low antigen densities (PMID: 31527070), we believe that the CD8 α hinge warrants consideration for future CAR Ts targeting pHLA neoantigens and other low antigen density targets. Another interesting aspect of our study was that the “improved” 2Q1.4 variant identified resulted in a more efficacious CAR T despite having reduced affinity towards the mutant (Fig. 6/5F/1E). Our results suggest that binding kinetics may be a better indicator of functional activity and should be investigated during the

design process and screening of variants. If a complex structure is available, modelling can help narrow down specific mutations to test in functional or binding assays; however, our results also highlight that predicting changes in affinity by computational methods remains challenging (e.g. seven of our eight mutations predicted to “improve” binding properties instead abrogated binding). To this Reviewer’s previous point, perhaps computational structural modelling can be employed alongside next generation sequencing-based CAR T library screening approaches to improve the hit rate of variant identification.

The discussion now highlights two elements of design for future neoantigen CARs:

-hinge

*“However, customization of the hinge region for CARs targeting MANA-pHLA complexes has yet to be explored. Here, we systematically screened four hinges with three neoantigen-specific TCRm scFvs targeting different HLA allele backgrounds (A*02:01, A*03:01, B*07:02) and identified the CD8α hinge as the most sensitive for our given target class of tumor-specific MANA-pHLA complexes.”*

-modelling to improve binding kinetics

We have modified the Discussion section to now include the following paragraph addendum:

“It is also worthwhile to note that while 2Q1.4 exhibited lower affinity than the parental 2Q1 clone, the 2Q1.4 CAR not only exhibited less off-target activity towards WT-peptide cells, but also increased reactivity towards mutant-peptide presenting cells. This may be explained by the kinetics of the interaction. Although 2Q1.4 has a lower affinity, its off-rate is 112-fold slower, resulting in a longer residency/dwell time and better functional potency. The on-rate, however, is three orders of magnitude slower (324-fold). One potential explanation for this is that the conformational ensemble of the parental Fab, 2Q1, is biased towards the bound conformation, and results in a faster association rate. The mutation of the heavy chain Y103 to histidine in 2Q1.4 either changes this conformational ensemble in such a way that no longer biases the bound conformation, or the bound conformation is now different. In turn, the more tightly associated pHLA-2Q1.4 Fab, results in a slower off-rate. These results would indicate that off-rate is a much better indicator of functional potency than binding affinity alone. Future studies incorporating unbiased high-throughput cell-based screening approaches, in tandem with computational structural modeling, could help in further clarifying the design parameters and improve the hit rate of variant identification.”

(3) It’d also be useful to elaborate the advantages of CAR-T for neoantigen targeting compared to TCR-engineered T cells or TCRm. With this structural insights and optimized binding moieties, why not just engineer and use TCR-T cells for treatment which after all are natural pMHC.

We thank this Reviewer for suggesting alternative therapeutic modalities as a way to target neoantigens. These various technologies have certainly not escaped our attention. While there may one day be a consensus on the optimal way to target neoantigens, at present multiple modalities are being investigated across various target indications, each with their inherent advantages and disadvantages. As this Reviewer has astutely pointed out, some of the most common protein- and cellular-engineering methods employed in targeting neoantigens and the like include TCR-CD3 bispecifics (PMID: 22561687), TCRm-CD3 bispecifics (PMID:

33649166), TCRm:TCR-T cells (PMID: 31064990), TCR-T cells (PMID: 26516200), and the TCRm:CAR T cell approach described herein. One obvious difference between TCRm antibodies versus native TCRs is the stark affinity differences between the two. However, TCRs that have undergone affinity-maturation to bring their affinity closer inline with TCRm antibodies have resulted in fatal toxicities (PMID: 23770775). Affinity aside, synthetic TCRm antibodies isolated from combinatorial phage display libraries allow for sampling beyond the scope and diversity of mouse and human TCR repertoires. While it remains to be determined which class of targeting moiety will prove most efficacious in humans, one significant advantage of our TCRm:CAR T cell approach (over the other four) is that it incorporates costimulation via CD28 and 4-1BB, a facet that has proved essential in the successful translation of CAR T cells to the clinic.

We have modified the Discussion section to now include the following paragraph addition:

“Various protein- and cellular-engineering methods have been employed to target neoantigens and the like, including TCR-CD3 bispecifics (PMID: 22561687), TCRm-CD3 bispecifics (PMID: 33649166), TCRm:TCR-T cells (PMID: 31064990), TCR-T cells (PMID: 26516200), and the TCRm:CAR T cell approach described herein. While it remains to be determined which class of therapeutic will prove most efficacious in humans, the TCRm:CAR T cell approach described in this work allows for the incorporation of antigen-dependent costimulation via CD28 and 4-1BB, a property that has proved essential in the successful translation of CAR T cells to the clinic.”

Reviewer #2 (Remarks to the Author):

Hwang et al report a very nice structure-function study on a clinically impactful problem. The authors have been very thorough in their analyses, with the crystal structures of pHLA binary complexes, the Fab unliganded, and the Fab ternary complex. The structures – which are documented well, are also supported by biophysical measurements. While the basis for the selectivity for the mutant epitope is somewhat obscure, the authors have provided some plausible explanations as to the mechanism. Often buried MHC polymorphisms, or HLA-buried viral escape mutants, can unexpectedly impact on TCR recognition, so perhaps the authors can reflect on this.

We thank for the reviewer for highlighting the opportunity to discuss polymorphisms. We have added the following, more in depth discussion to the paper:

*“To deepen the understanding of 2Q1 recognition, we aligned and performed subsequent structural analysis of the HLA-B allomorphs, focusing on polymorphic residues that could affect peptide-MHC binding and recognition by the TCRm antibody. For example, buried Tyr116 in HLA-B*07:02 (Asp116 in HLA-B*27:01, Phe116 in HLA-B*35:01) which is at the β -sheet base, is within hydrogen bonding distance of IDH2^{R140Q} the neoantigen displayed. This would significantly affect binding of the neoantigen to the HLA. Another polymorphic residue worth mentioning is Asn63 in HLA-B*07:02 (Glu63 in HLA-B*27:01 and HLA-B*13:01) since the glutamate polymorphism would clash with the proline anchor residue at position 2 of the IDH2^{R140Q} peptide.”*⁶⁴

Buried viral escape mutants result in drastic changes in the overall structure or conformation of the peptide-bound in the MHC in turn leading to T cell escape (PMID: 24759101, 28218747). However, in the case of the IDH2^{R140Q} buried epitope bound to the HLA, there is not a drastic conformational change in the overall structure of the peptide compared with the IDH2^{WT} peptide. Nevertheless, the small subtle changes observed are enough to affect recognition and binding affinity.

We have added the following reference to viral escape mutants in discussion:

“This is in contrast to known buried viral escape mutants which result in peptide bulging.”

I note that the mutant results in a slower off rate (Fig. 1D vs 1E), but the authors do not speculate why.

The mutant does indeed result in a slower off-rate, resulting in a longer residency time, which may contribute to its improved functional potency in ELISAs and when grafted into a CAR-T cell format. The on-rate is significantly slower (about 3 orders of magnitude), which is the largest contributor to the lower affinity. One potential explanation for this is that the conformational ensemble of the parental Fab, 2Q1, is biased towards the bound conformation, and results in a faster association rate. The mutation of heavy chain Y103 to histidine, either changes this conformational ensemble in such a way that no longer biases the bound conformation, or the bound conformation is now different and results from a less frequent conformation. The resultant bound conformation, however, is a tighter association and thus results in a slower off-rate.

We have added the following paragraph in the discussion on pg. 17 to expand on this.

“It is also worthwhile to note that while 2Q1.4 exhibited lower affinity than the parental 2Q1 clone, the 2Q1.4 CAR not only exhibited less off-target activity towards WT-peptide cells, but also increased reactivity towards mutant-peptide presenting cells. This may be explained by the kinetics of the interaction. Although 2Q1.4 has a lower affinity, its off-rate is 112-fold slower, resulting in a longer residency time and better functional potency. The on-rate, however, is three orders of magnitude slower (324-fold). One potential explanation for this is that the conformational ensemble of the parental Fab, 2Q1, is biased towards the bound conformation, and results in a faster association rate. The mutation of heavy chain Y103 to histidine in 2Q1.4, either changes this conformational ensemble in such a way that no longer biases the bound conformation, or the bound conformation is now different. In turn, the more tightly associated pHLA-2Q1.4 Fab, results in a slower off-rate. These results would indicate that off-rate is a much better indicator of functional potency than binding affinity alone⁷⁶.”

Did authors look at B-factors of the binary complexes?

We appreciate the opportunity to discuss the qualities of the data. Although each peptide in both structures have good, low B-factors, (mutant IDH2^{R140Q}-HLA-B*07:02 has 34 Å² and the IDH2^{WT} has 40 Å²), we think the small difference in B-factor is correlated with the resolution of the structures (mutant IDH2^{R140Q} 1.9 Å vs IDH2^{WT} 2.25 Å). In structures determined by X-ray crystallography the B-factor ‘collects’ the errors in many aspects of the rebuilding and

refinement such as disorder and will not necessarily correlate with thermal stability of the binary complexes.

The manuscript was modified to include this information in the caption of Fig. 2.

Examining thermal stability of the pHLA binary complexes could be informative.

We appreciate the suggestion of adding the experimental measurement. We have determined the thermal stability of both the IDH2^{WT} and IDH2^{R140Q} peptide bound HLA-B*07 complexes, and this has been added as Supplementary Figure 3. The mutation of Arg140 to glutamine results in a 5° decrease in thermal stability of the pHLA complex. This is presumably due to the loss of salt-bridge with Asp114, and the loss of π - π interactions with Tyr99, as discussed on page 9.

We have added the sentence to the manuscript:

“This loss of interaction result in a 5° decrease in the melting temperature of the pHLA complex.”

It’s a comprehensive, well-written and nicely presented paper that requires limited modifications.

Minor comments:

SPR experiments. N=? independent measurements

Thank you for bringing this omission to our attention. The SPR experiments were done in duplicates (N=2). This information was added in the caption of Fig. 1D & E and Fig. 5E & F.

As the ternary complex was solved at modest resolution, the structural differences observed could just be due to coordinate error. Authors should acknowledge this

We thank the reviewer for the opportunity to clarify the extent of the structural differences. We have added the following sentence in the section “Structural determinants of 2Q1 selectivity for IDH2”

“Despite the limited resolution of the data, clear electron density was observed at the binding interface for the IDH2^{R140Q} peptide and the CDRs of the 2Q1-Fab (fig. S4E).”

We have also elaborated on the error of the coordinates with the following sentence in page 11:

“The 1.2 Å shift in the coordinates of the carbonyl oxygen atoms between IDH2^{R140Q} in the ternary complex (PDB ID 6UJ9, 2.9 Å) and IDH2^{WT} in the WT pHLA structure (PDB ID 6UJ8, 2.25 Å), albeit small, is about three-times the estimated coordinate error of 0.41 and 0.2 Å, respectively.”

ref 48 is incorrect wrt TCR-pMHC structures.

Høydahl, L. S., Frick, R., Sandlie, I. & Løset, G. Å. Targeting the MHC Ligandome by Use of TCR-Like Antibodies. *Antibodies* 8, 32 (2019).

We are not entirely sure what the reviewer means by this comment. Figure 1 in ref.48 (Høydahl et al (2019)), overlays published TCR:pHLA crystal structures, as well as TCRm:pHLA crystal structures and highlights the conserved binding mode of TCRs, in contrast to the diversity in binding modes of TCRm. As such, we believe this reference is appropriately placed following

both the sentence describing the canonical TCR/pHLA binding mode, as well as the diversity of orientations of TCRm/pHLA complexes. Upon further examination, we interpreted this correction as that reference 49 was incorrectly placed in the following sentence and it has been relocated to the preceding sentence.

“For example, crystal structures of TCR-pHLA complexes generally favor a canonical diagonal binding mode, where the TCR V α complementarity determining regions (CDRs) sit atop the N-terminus of the peptide and the V β CDRs over the C-terminus^{48,49.}”

“The growing number of TCRm antibody structures bound to pHLA complexes, however, are beginning to reveal a wide-range of orientations, as measured by docking and incident angles⁴⁸”

Any comments/thoughts why the Fab is specific for the B*07:02 allomorph vs other HLA molecules?

We thank for the reviewer for the opportunity to discuss the specificity in light of allomorphs.

We have added the following to the result section:

*“NetMHCpan, which predicts peptide binding to HLA alleles, estimates that the mutant IDH2^{R140Q} peptide will bind 200-fold tighter to the HLA-B*07:02 allomorph compared with other common HLA alleles.”*

This peptide specificity, along with the extensive peptide and HLA contacts that the antibody makes, will be responsible for the specificity of the Fab for the B*07:02 allomorph. The paragraph added in reference to the Reviewer’s earlier comment (see below) on buried polymorphisms also goes towards explaining why the Fab is specific for the B*07:02 allomorph.

*“To deepen the understanding of 2Q1 recognition, we aligned and performed subsequent structural analysis of the HLA-B allomorphs, focusing on polymorphic residues that could affect peptide-MHC binding and recognition by the TCRm antibody. For example, buried Tyr116 in HLA-B*07:02 (Asp116 in HLA-B*27:01, Phe116 in HLA-B*35:01) which is at the β -sheet base, is at hydrogen bonding distance of IDH2^{R140Q}, the neoantigen displayed. This would significantly affect binding of the neoantigen to the HLA. Another polymorphic residue worth mentioning is Asn63 in HLA-B*07:02 (Glu63 in HLA-B*27:01 and HLA-B*13:01) since the glutamate polymorphism would clash with the proline anchor residue at position 2 of the IDH2^{R140Q} peptide⁶⁴.”*

Reviewer #3 (Remarks to the Author):

This interesting study reports the development of pHLA-targeting, TCR-mimicking CARs that recognize the IDH2 R140Q mutation presented by HLA-B*0702. Phage display with positive selection with the mutant and negative selection with the wildtype pHLA was used to generate several scFV clones, of which one (‘2Q1’) was characterized in detail biophysically and structurally. Surprisingly, the mutant and wildtype pHLA complex structures are highly similar, with the mutated position facing inward toward the MHC and only very minor backbone

differences between the two peptides. Nonetheless, the 2Q1 scFV is able to discriminate between the two with reasonable specificity (but still some binding to the WT pHLA), in part through a side chain-backbone hydrogen bond from R102(H3) to the carbonyl oxygen of 140Q. Structural modeling is then used to identify a mutation in 2Q1 (Y103H) that improves the specificity for the mutant by weakening binding to the wild type pHLA (and perhaps also to the mutant, but not by as much). Overall, there are a number of interesting findings here and the study seems well worth publishing. The observation of such strong specificity at an inward-facing peptide position is an important finding, which together with the Y103H mutation highlights once again the subtlety of macromolecular interactions.

I have a few minor questions/suggestions, as follows.

When I compare the mutant and wildtype peptide backbones in Figure 2F it does look like the mutant sits a little higher and perhaps has the carbonyl oxygen angled slightly differently. But it's difficult to tell from the angle. Could the wildtype peptide from the pMHC structure be superimposed onto the mutant peptide in the 2Q1 co-complex structure, to see if the R102 Hbond could still be made? This would not be at all conclusive but I'm curious whether it would provide any insight. Of course there is ample room for structural relaxation here.

We appreciate the suggestion to add a more detailed figure that highlight the structural differences.

The alignment of both peptides is in Fig. 3D; to ease the appreciation of the difference we have added a supplementary figure (Fig. S5) that zooms into the carbonyl oxygen of IDH2¹⁴⁰ to show the subtle difference.

The description of the structural modeling is quite brief: how were the mutations selected? On the basis of predicted binding affinities? More details would be useful here, even if the answer is that it was mainly human intuition in the end.

The final mutations were selected primarily based on their predicted binding affinities. We have added the comment '*based on their predicted binding affinities*' to the text (pg. 12). We have also updated a typo in the original text, '*from 5 Å to 8 Å of the epitope residue of interest*'. Moreover, we added a figure (fig. S7A) that shows the residues selected to be mutated in the 2Q1 (A32, W101, R102, Y103).

The sequence logos in Fig 5C, D do not look right at all. Traditionally, the height of a letter stack in this type of visualization is related to the degree of conservation at that position. So for example logos of allele-specific, Class I-presented peptides typically show tall columns at position 2 and N, the anchor positions. I don't know how the flow data was converted into sequence logos but it does not seem to match one's expectations. For example, the position 2 stack is very small, even though the heat map to the left suggests high specificity. Similarly at position 7, the heat map seems to suggest that 2Q1.4 is more specific. But the letter stack is higher for 2Q1. Please first of all clarify how the heatmaps are turned into logos, and consider using a more intuitive algorithm so that stack height is indicative of specificity.

We thank the reviewer for the helpful comment. Indeed, our sequence logos graph is different from the conventional ones and this is due to the difference in the source data. In conventional sequence logos, such as the MHC binding motifs described by the reviewer, the input data represents the “frequency” of a certain amino acid at each position, calculated from a large number of binding peptides. Hence the height of the characters reflects their relative frequency. In our case, the input data was the “intensity” by which our antibodies stain each variant peptide. We transformed the heatmap into logos by first dividing the “median fluorescence intensity (MFI)” of each peptide by 100,000, and used the PSSM-logo algorithm. In our case, the size of individual letters reflects staining intensity. This method of generating logos has been described before to indicate the binding affinity of peptide variants to an HLA⁸⁷.

Alternatively, we could artificially transform the staining intensity into a “pseudo-frequency,” in which we divide the MFI of a particular variant peptide by the sum of the MFIs of all the 20 peptides with amino acid variation at the same position. The visual output from this method is not as intuitive (see below).

2Q1 IgG1

2Q1 IgG1.4

We have added in methods the following:

“The Seq2Logo graph was generated as described previously to indicate the binding affinity of peptide variants on an HLA⁸⁷. Specifically, it was calculated by dividing the “median fluorescence intensity (MFI)” of each peptide by 100,000, and using it as input for the PSSM-logo algorithm. The size of the individual letters reflects the staining intensity. “

REVIEWERS' COMMENTS

Reviewer #1 (Remarks to the Author):

I am delighted to see my comments have been adequately addressed.

Reviewer #2 (Remarks to the Author):

The authors have addressed my comments well, and I congratulate them on a fine study

Reviewer #3 (Remarks to the Author):

Overall my concerns have been addressed.

I do continue to disagree with the authors about how to display their MFI data as a logo. I checked their proposed reference (#87), and sure enough those folks are also doing this incorrectly. Or at any rate, in a way that is inconsistent with established usage. Again the key point here is that where the heights of the columns are variable, the tallest columns are the ones that have the most "information content" / correspond to the positions with the greatest specificity. As a thought experiment, consider a position where mutations have no effect on binding. In the authors' approach, this will be the tallest column in the logo. Indeed, I much prefer the pseudo-frequency logos that the authors generated in response to my original suggestion. In those, the increased specificity of the 2Q1.4 IgG1 at positions 3, 5, and 7 is readily apparent. The only thing left to do is to scale these by information content, which is a standard option and should be available through the Seq2logo website. But don't take my word for it! Ask anyone who thinks about sequence logos a lot and I imagine they will agree.

Answers to REVIEWERS' COMMENTS

Reviewer #1 (Remarks to the Author):

I am delighted to see my comments have been adequately addressed.

We thank this Reviewer for all their helpful comments—we were delighted to incorporate their suggestions.

Reviewer #2 (Remarks to the Author):

The authors have addressed my comments well, and I congratulate them on a fine study

We thank this Reviewer for all their helpful comments—we were delighted to incorporate their suggestions and appreciate this Reviewer's kind sentiments on our study.

Reviewer #3 (Remarks to the Author):

Overall my concerns have been addressed.

I do continue to disagree with the authors about how to display their MFI data as a logo. I checked their proposed reference (#87), and sure enough those folks are also doing this incorrectly. Or at any rate, in a way that is inconsistent with established usage. Again the key point here is that where the heights of the columns are variable, the tallest columns are the ones that have the most "information content" / correspond to the positions with the greatest specificity. As a thought experiment, consider a position where mutations have no effect on binding. In the authors' approach, this will be the tallest column in the logo. Indeed, I much prefer the pseudo-frequency logos that the authors generated in response to my original suggestion. In those, the increased specificity of the 2Q1.4 IgG1 at positions 3, 5, and 7 is readily apparent. The only thing left to do is to scale these by information content, which is a standard option and should be available through the Seq2logo website. But don't take my word for it! Ask anyone who thinks about sequence logos a lot and I imagine they will agree.

We thank this Reviewer for their suggestion to conform with the established usage of the logos. We have now replaced the MFI data as a Seq2logo with the tallest columns corresponding to the positions with the greatest specificity in Figure 5.